# Structural snapshots of *Pseudomonas aeruginosa* LptB$_2$FG and LptB$_2$FGC reveal insights into lipopolysaccharide recognition and transport

Francesco Fiorentino [1,2], Matteo Cervoni [3], Yi Wang[4], Leonhard H. Urner [5], Joshua B. Sauer[6], Anh Tran[7], Robin A. Corey [8], Dante Rotili[3], Antonello Mai [9], Phillip J. Stansfeld [10], Francesco Imperi[3], Edward W. Yu [7], Chih-Chia Su [7] ✉, Carol V. Robinson [2,11] ✉ & Jani R. Bolla [4,11] ✉

Gram-negative bacteria are intrinsically resistant to many antibiotics because of densely packed lipopolysaccharides (LPS) in the outer leaflet of their outer membrane (OM), which acts as a highly effective barrier towards the spontaneous permeation of toxic molecules, including antibiotics. LPS are extracted from the inner membrane by the ABC transporter LptB$_2$FGC and translocated across the periplasm via a protein bridge to the OM. While structural studies have elucidated aspects of Lpt function in enterobacteria, little is known about how this system operates in divergent species such as *Pseudomonas aeruginosa*, a major human pathogen. Here, we report five cryo-electron microscopy structures of *P. aeruginosa* LptB$_2$FG and LptB$_2$FGC, revealing a rigid body movement in the periplasmic β-jellyroll domains necessary for LPS to shuttle through the periplasmic space. Notably, these structures exhibit a significantly smaller LPS binding cavity compared to previously determined models, suggesting the ligand-unbound states of the transporter. Mass spectrometry and molecular dynamics simulations indicate that the phosphate groups of LPS are the key determinants for binding and that the transporter can also accommodate cardiolipin. Together, these findings reveal previously unappreciated structural diversity in the Lpt system and provide mechanistic insight into how pathogenic Gram-negative bacteria tailor LPS recognition and transport. This understanding offers new avenues for the development of novel inhibitors targeting membrane biogenesis.

Infections caused by Gram-negative bacteria are difficult to treat in part, owing to the low permeability of the outer membrane (OM) to antibiotics. Consequently, few antibiotics are active against Gram-[1].negative bacteria, and multidrug-resistant strains are rising[2].

The inner membrane (IM) of Gram-negative bacteria is mainly composed of phospholipids, while the OM is an asymmetric bilayer presenting lipopolysaccharides (LPS) on the outer leaflet and phospholipids in the inner leaflet[3]. The amphiphilic nature of LPS

guarantees bacterial protection towards both polar molecules and toxic hydrophobic compounds, such as antibiotics, and is essential for stabilising the overall OM structure[4,5].

LPS is a glycolipid consisting of lipid A, core oligosaccharide and O-antigen polysaccharide. Mature LPS is extracted from the outer leaflet of the IM and transported through the periplasm to be inserted into the outer leaflet of the OM by seven essential proteins LptA-G (Fig. 1a)[6-8]. The IM sub-complex comprises the ATP-binding cassette (ABC) transporter LptB$_2$FGC, which extracts LPS from the IM and delivers it to the periplasmic components of the Lpt system (Fig. 1a)[9,10]. The periplasmic protein LptA (named LptH in *Pseudomonas aeruginosa*) forms a continuous bridge with LptC and LptD β-jellyroll domains spanning across the periplasm[11-13]. Finally, the heterodimer LptDE inserts LPS into the outer leaflet of the OM (Fig. 1a)[14-18]. In some Gram-negative bacteria, additional lipoproteins contribute to LPS transport efficiency and system stabilisation. For example, LptM has been proposed to act as an accessory factor that stabilises LptD assembly or modulates transport under stress conditions[19,20]. More recently, the lipocalin YedD was identified as a component of the translocon and is required for optimal LPS transport[21].

LptB$_2$FGC is a type VI ABC transporter consisting of 5 subunits[22]. The LptB cytoplasmic subunits form the nucleotide-binding domains (NBDs) that capture and hydrolyse ATP. LptF and LptG possess transmembrane helices that create a cavity and a translocation path for LPS, as well as a periplasmic β-jellyroll domain. Differently from other ABC transporters, this sub-complex comprises the single-pass membrane protein LptC with its transmembrane helix situated between the two transmembrane domains (TMDs) of LptF and LptG and plays an important role in regulating the ATPase activity to allow higher LPS extraction efficiency of the protein complex[23-27]. LptC also contains a β-jellyroll domain, which is in contact with LptF and LptA in the periplasm to form a periplasmic bridge. Structural data for LptB$_2$FGC have largely been derived from enterobacteria such as *Escherichia coli*, *Klebsiella pneumoniae*, *Enterobacter cloacae*, and *Shigella flexneri*, with only two more divergent homologues characterised in *Vibrio cholerae* and *Acinetobacter baylyi*[23,28-31]. In these complexes, the periplasmic β-jellyroll domains have been resolved only in *E. cloacae* and *V. cholerae*, where LptC interacts directly with LptF to form a simple heterodimer, while LptG appears to act primarily as a structural scaffold[23].

Genetic studies have shown that *P. aeruginosa* LptC (*Pa*-LptC) cannot functionally substitute *E.coli* LptC, likely due to a lack of assembly with the *Ec*-LptB$_2$FG complex[32]; *Pa*-LptC shares only ~20% sequence identity with its *E. coli* counterpart, highlighting the divergence between these two Lpt systems. Therefore, which β-jellyroll of LptF and LptG partners with *Pa*-LptC and serves as the LPS exit pathway, has been unclear. The first crystal structure of *Pa*-LptB$_2$FG[33] suggested that either LptF or LptG could both serve as the lateral LPS exit route. This prompted the proposal of an alternating lateral access model[34], later discounted by cryo-EM structures that confirmed LptF as the primary LPS exit route[23,28].

Previous structural investigations using X-ray crystallography and cryo-EM for LptB$_2$FG and LptB$_2$FGC have successfully unveiled the proposed mechanism for the structural basis of LPS extraction by LptB$_2$FGC[23-25,31]. Nevertheless, several unresolved questions persist regarding the LptB$_2$FGC transporter. The true structure of its apo state remains unclear, since all previously published cryo-EM structures contain LPS, while X-ray structures present similarly sized cavities, although LPS-free[23-25,35]. This raise concerns as to whether these X-ray structures truly represent apo forms. In addition, the range of configurations adopted by the β-jellyroll domains throughout the LPS transport cycle is not fully understood. The mechanisms that enable LPS to enter the transmembrane binding cavity, particularly when obstructed by the transmembrane helix of LptC, also remain elusive. Moreover, it is uncertain how selective the transporter is for LPS or whether it might recognise other membrane lipids. Finally, it is not clear how LptF and LptG might contribute differently to transport in divergent Gram-negative bacteria.

In the present study, we have employed an integrated approach to address these unresolved issues. Guided by native mass spectrometry (MS), we optimised and purified an LPS-free LptB$_2$FG(C) transporter of *P. aeruginosa*. Then, using cryo-EM, we uncovered three major structural populations of LptB$_2$FG and two major populations of LptB$_2$FGC featuring a rigid body movement in the β-jellyroll domains. Further comparison with existing structures indicated that our structures represent the ligand unbound states of the transporter. We further used native MS to probe the structure-activity relationship of LPS and studied how membrane lipids such as cardiolipin influence LPS binding. These results, supported by molecular dynamics simulations, reveal critical residues involved in lipid recognition. Our data provide

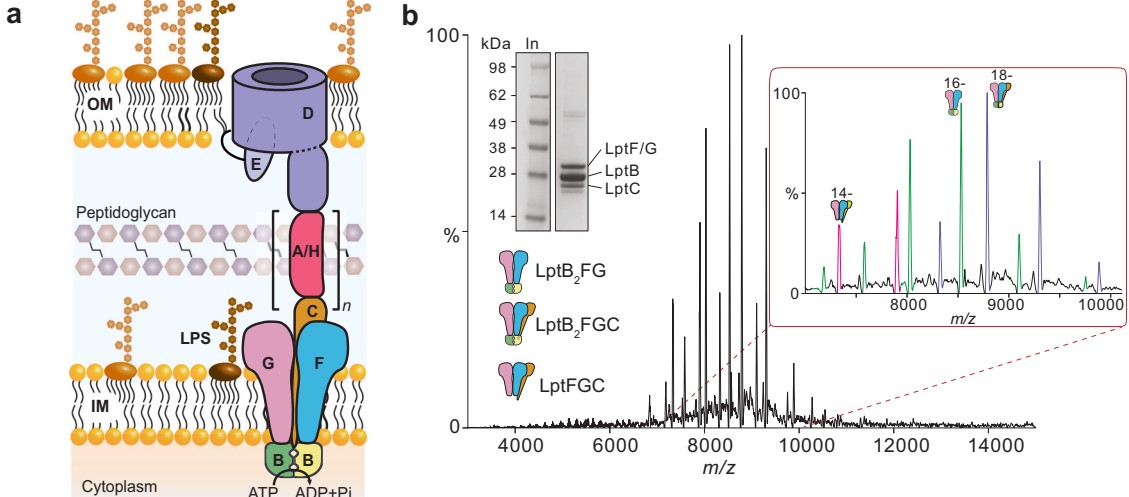

**Fig. 1 | Purification of LPS-free LptB$_2$FG/LptB$_2$FGC. a** Cartoon representation of the LPS transport system. **b** Native mass spectra of the delipidated LptB$_2$FG/LptB$_2$FGC complex, indicating that both complexes are present in approximately equal quantities. Inset SDS-PAGE gel of the purified complex. The gel was run under non-reducing conditions. The bands corresponding to LptB/F/G/C are clearly seen

at the expected size; the ~52 kDa band is the LptB dimer, and the band below LptC corresponds to the common contaminant *E. coli* protein (CRP). The data shown are representative of three independent experiments, each performed with biological replicates (*n* = 3).

new mechanistic insights into LPS transport and expand the structural landscape of LptB$_2$FGC complexes from non-enterobacterial species. Our findings identify functionally important distinctions in the *P. aeruginosa* LptB$_2$FGC complex, provide a structural explanation for the previously observed genetic incompatibility, and highlight opportunities for species-specific antibiotic development.

## Results

### Purification of LPS-free LptB$_2$FG(C)

We initially expressed *P. aeruginosa* LptB$_2$FGC in the *Escherichia coli* strain BL21(DE3) and purified this protein complex following a classic two-step purification protocol consisting of immobilised metal affinity chromatography (IMAC) and size-exclusion chromatography (SEC), as indicated from previous studies[23–25,31,35]. The protein complex was initially purified in a buffer containing 0.03% (w/v) n-dodecyl-β-D-maltopyranoside (DDM), following a delipidation protocol to remove bound lipids, especially LPS[36,37]. We then performed a detergent screen to establish the ideal conditions for native MS analysis[38]. Among the tested detergents, the recently reported [G1], dendritic oligoglycerol detergent ([G1]-OGD) showed an ability to retain non-covalent interactions and allowed for facile detergent removal from proteomicelles, thus yielding well-resolved spectra[39]. Native mass spectra in [G1]-OGD revealed charge states corresponding to both LptB$_2$FG and LptB$_2$FGC complexes, which may be due to either an LptB$_2$FG/LptB$_2$FGC equilibrium or lower expression levels of LptC compared to other subunits. Moreover, we observed adducts ranging from 1.5 to 4.7 kDa, which are consistent with co-purified phospholipids (PLs) and LPS[40,41] (Supplementary Fig. S1). Overall, the presence of these adducts indicates the potential absence of a truly apo form of the LptB$_2$FG and LptB$_2$FGC complexes from previous studies, which could explain why the LPS binding cavities in the X-ray and cryo-EM structures are of the same size.

In the quest for an in-depth understanding of the mechanism underpinning LptB$_2$FGC-mediated LPS extraction from the outer leaflet of the IM, we aimed to obtain LPS-free LptB$_2$FG and LptB$_2$FGC complex structures. Hence, we overexpressed the protein complex in ClearColi BL21(DE3) cells[42]. ClearColi (genotype: *ΔgutQ, ΔkdsD, ΔlpxL, ΔlpxM, ΔpagP, ΔlpxP, ΔeptA, msbA148*) cells are able to produce only lipid IVA, in place of the characteristic hexa-acylated enterobacterial lipid A, as the only LPS-related molecule in the outer membrane. We purified the protein, expressed in ClearColi, in DDM following the same procedure as described above. We confirmed with SDS-PAGE that LptC was successfully co-expressed with the other complex subunits (Fig. 1b, inset). Next, we carried out native MS experiments in [G1]-OGD, which showed the presence of charge states corresponding to both LptB$_2$FG and LptB$_2$FGC (Fig. 1b). Notably, no lipid-bound species were detected, indicating that we were able to successfully remove lipid IVA or any other membrane lipids potentially associated with the transporter during expression in ClearColi cells. As a result, we purified the ligand-unbound forms of the two protein complexes (see Supplementary Fig. S1 for comparison) (Fig. 1b). With the improved resolution of the mass spectra, we were able to determine the relative abundance of the two protein complexes and estimate a 1:1 ratio between LptB$_2$FG and LptB$_2$FGC.

### Purified LptB$_2$FG(C) has ATPase activity

To further confirm that the purified LptB$_2$FG(C) is active, we performed an ATPase assay in both purification and MS buffer, and the values obtained are in line with previously reported ones, verifying that the complexes we obtained are in their active forms[25] (Supplementary Fig. S2).

In addition, previous reports indicated that tagging LptB at the C-terminus compromises its function in *E. coli*[43] but retains ATPase activity in vitro. To assess this possibility for *P. aeruginosa*, we performed a functional assay using an *lptB* conditional mutant (PAO1

*rhaSR-PrhaBAD-lptB ΔlptB*)[44] transformed with plasmids that express LptB of *P. aeruginosa* PAO1 or the LptB variant used in this study with or without the C-terminal 6 × His tag. Rhamnose and IPTG were used to induce the chromosomal copy of *lptB* or the *lptB* allele cloned into the plasmid pME6032, respectively. Functional assays demonstrated that the His-tagged variants successfully complemented the *lptB* conditional mutant, even though the colonies were slightly smaller than those of the control strains expressing non-tagged LptB. This indicates that the tagged LptB variants are at least partially functional in vivo in *P. aeruginosa* (Supplementary Fig. S3). Together, these findings confirm that our purified LptB$_2$FG(C) complexes are enzymatically competent in vitro and that the tagged LptB variants support growth in complementation assays, and we therefore used these preparations for the structural and biochemical analyses reported here.

### Substrate-free structures of LptB$_2$FG in lipid nanodiscs

Having succeeded with the purification of functional apo states of LptB$_2$FG/LptB$_2$FGC, we reconstituted the protein complexes into lipid nanodiscs and collected single-particle images using cryo-EM (Supplementary Figs. S4, S5, S6, and S7). We then processed this cryo-EM data using the Build and Retrieve (BaR) methodology, following the steps described previously[45]. Extensive classification of the single-particle images indicated there are several distinct populations of LptB$_2$FG and LptB$_2$FGC with various conformations coexisting in the cryo-EM sample. Several iterative rounds of classifications allowed us to sort the images based on five distinct conformations, which correspond to three different LptB$_2$FG and two different LptB$_2$FGC states (Supplementary Fig. S4 and S5).

Three distinct conformations of LptB$_2$FG, designated as LptB$_2$FG-I, LptB$_2$FG-II, and LptB$_2$FG-III, were captured in our cryo-EM data and solved to resolutions of 3.34 Å, 3.26 Å, and 3.31 Å, respectively (Supplementary Table S1, Fig. 2a and Supplementary Fig. S8). Overall, all the cryo-EM structures of LptB$_2$FG adopt the fold of the previously solved LptB$_2$FG structure[23–25,31,35], in which two copies of LptB participate in forming the NBDs, whereas LptG and LptF are engaged in creating the TMDs made up of 12 transmembrane helices and the periplasmic β-jellyroll domains. The main difference between the three cryo-EM structures of LptB$_2$FG is in the conformation of the β-jellyroll domain, which appears to perform a rigid body rotation motion with respect to the outer leaflet of the IM, transitioning from one conformational state to the other (Fig. 2b). The TMD and NBD domains of these three structures are essentially the same. The transmembrane helices create an outward-open cavity with two interfaces: Interface 1, formed by the interaction between TM1$^{LptG}$ and TM5$^{LptF}$, and Interface 2, established by the connection between TM1$^{LptF}$ and TM5$^{LptG}$ (Supplementary Fig. S9a).

In contrast to all the reported structures of LptB$_2$FG determined by cryo-EM, our cryo-EM structures did not reveal any density consistent with the presence of LPS within the outward-facing cavity of the TMDs (Fig. 2c). This cavity has a volume of 5,473 Å$^3$, approximately half the volume observed in the X-ray structure (10,513 Å$^3$, Supplementary Fig. S9b), suggesting that the previously solved structure may represent an LPS-bound conformation. Intriguingly, our structures of LptB$_2$FG exhibit an open V-shaped cleft at Interface 1, oriented toward the outer leaflet of the inner membrane, while Interface 2 remains entirely closed (Fig. 2d). This structural feature is expected to facilitate lipids entering the TMD domain of LptB$_2$FGC via Interface 1. These observations are in line with biochemical and crosslinking investigations that establish Interface 1 as the point of LPS entry[23–25] (Supplementary Fig. S10). The distinctive structural attributes characterising our cryo-EM LptB$_2$FG structures strongly suggest that we have captured the resting state, or substrate-free conformation, of this transporter.

The open and closed states of interfaces 1 and 2 likely involve rotations of two rigid-body groups within the TMD of LptF

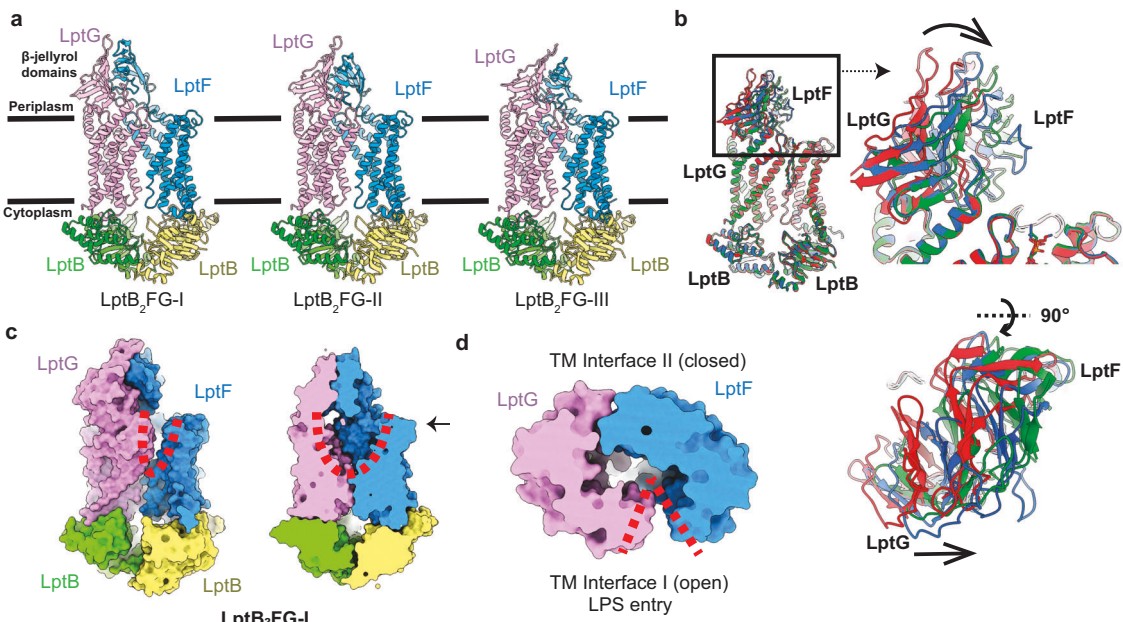

**Fig. 2 | Substrate-free structures of LptB$_2$FG. a** Cryo-EM structures of LptB$_2$FG in three different conformations. **b** Structural superposition of three conformational states of the LptB$_2$FG complex: form I (red), form II (blue), and form III (green). The overlay reveals conformational differences primarily localised to the periplasmic β-jellyroll domains of LptF and LptG. Close-up side view of the β-jellyroll domains in LptF and LptG demonstrates a rigid-body rotation, predominantly clockwise, toward the outer leaflet of the inner membrane (right). Top-down view of the β-

jellyroll domains, highlighting the direction of conformational shifts across the three forms (below). The overlay of three structures suggests rigid-body movement in the β-jellyroll domains. **c** cross-section at the top of the transmembrane helix domain showing the binding cavity of LptB$_2$FG responsible for LPS binding. **d** The top-down view of LptB$_2$FG-I shows that Interface 1 is open while Interface 2 is closed in these structures.

(Supplementary Fig. S9). The movement of the first rigid-body group (TM1-3) would be linked to the opening and closing of interface 2, while the second rigid-body group (TM4-5) would regulate access to interface 1. In our substrate-free LptB$_2$FG structures, both rigid groups 1 and 2 rotate counterclockwise toward LptG (viewed from the top of the transmembrane domain), causing group 2 to move away from LptG, thereby opening interface 1. Simultaneously, group 1 moves toward LptG, closing Interface 2 and reducing the volume of the transmembrane domain binding cavity.

We identified four lipid molecules in LptB$_2$FG-II (Fig. S8a, b) and LptB$_2$FG-III (Supplementary Fig. S8d), likely originating from the nanodisc preparation used for sample reconstitution. These lipids were modelled as phosphatidylethanolamine (PE) moieties, arranged on the outer surface of the LptB$_2$FG complex. Among these lipids, PE1 (Supplementary Fig. S8a) was located outside of interface 1, at the level of the outer leaflet of the cytoplasmic membrane. The other three lipids were positioned in the inner leaflet, with PE2 located in the vicinity of interface 1 (Supplementary Fig. S8a), and PE2 and PE3 at interface 2 (Supplementary Fig. S8b). In the LptB$_2$FG-I structure, we observed PE1, PE3, and PE4 (Supplementary Fig. S8c).

### Structures of resting conformation of LptB$_2$FGC in lipid nanodiscs

We also obtained two cryo-EM 3D reconstitutions of LptB$_2$FGC at resolutions of 3.26 Å and 3.61 Å, respectively (Fig. 3a, Supplementary Table S1, Supplementary Figs. S4, S5, S7 and S11). These two structures were assigned as LptB$_2$FGC-I and LptB$_2$FGC-II. We then refined the cryo-EM structure of these two complexes. Structure alignment of these two structures gives rise to a root mean square deviation (RMSD) of 1.2 Å over 1324 Cα atoms and indicates the main difference between these two cryo-EM structures is in the β-jellyroll structures. These regions appear to perform a rigid body rotation motion toward the outer leaflet of IM (Fig. 3b), similarly to what we observed for LptB$_2$FG (Fig. 2b). The TMD and NBD domains of LptB$_2$FGC are

similar to those of the LptB$_2$FG structures and represent the resting state. In our LptB$_2$FGC structures, the periplasmic β-jellyroll of LptC was clearly resolved and found to form a trimeric interface: LptC and LptF form the primary LPS transfer path, while a loop from LptG (residues 209–215) binds onto the N-terminal β-jellyroll of LptC (residues 60–69) (Fig. 3c). In contrast, in the LptB$_2$FGC structure from *Vibrio cholerae* (6MJP), LptC interacts predominantly with LptF, and LptG makes only minimal contact with LptC (Supplementary Fig. S12), resulting in an interface that resembles a heterodimer rather than a heterotrimer.

Intriguingly, in contrast to previously documented LptB$_2$FGC structures (Supplementary Fig. S13), the transmembrane helix of LptC remained unobservable in these maps, indicating its flexibility and a lack of engagement with LptF and LptG at Interface 1. This, in turn, results in an unobstructed entry point for LPS into the TMD cavity. Similar to LptB$_2$FG, we identified four lipids in the transmembrane region of the form I conformation, which likely originate from nano-disc reconstitution (Supplementary Fig. S11a, b). However, for LptB$_2$FGC-II, the lower resolution and particle counts in this conformation prevented confident lipid identification. A comparison of our cryo-EM structure of LptB$_2$FGC-I with the previously reported *V. cholerae* LptB$_2$FGC (6MJP) reveals a significant conformational difference (RMSD 5.57 Å across 845 Cα atoms), which may reflect distinct functional states.

### LPS binding to LptB$_2$FG

To test whether the purified *P. aeruginosa* LptB$_2$FG(C) complex could still bind LPS after delipidation, we tested the binding in vitro with exogenously added LPS. We performed native MS using both *P. aeruginosa* LPS (*Pa*-LPS) and *E. coli* LPS (*Ec*-LPS) at defined concentrations to evaluate binding preferences and specificity. Since LPS is a heterogeneous mixture[46] and to avoid the complications in interpreting our native MS analysis, we have purified *P. aeruginosa* LptB$_2$FG (without LptC) using our established methodology from above.

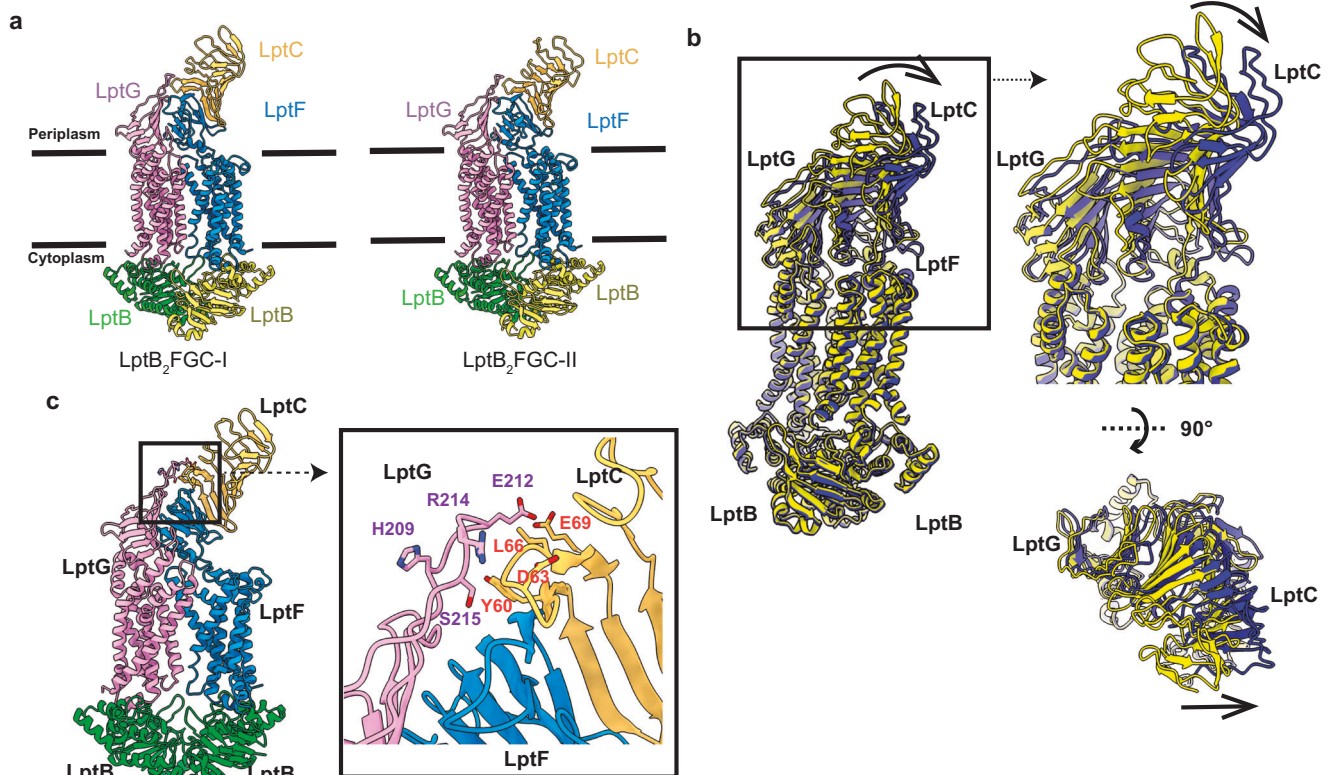

**Fig. 3 | Substrate-free state structures of LptB₂FGC. a** Cryo-EM structure of LptB₂FGC showing two different conformations. **b** Overlay of the LptB2FGC structures, showing rigid body movements in the periplasmic portions of the complex. Form I is shown in yellow and form II in purple, highlighting the conformational changes in the periplasmic regions of the transporter. (right) Zoomed-in view of the β-jellyroll domains, illustrating clockwise rotation toward the outer leaflet of the inner membrane. (bottom) Top view of the β-jellyroll domains, showcasing the overall conformational movement. **c** Overall structure of the LptB₂FGC complex, highlighting the interaction between LptC and LptG. A loop region in LptG (residues 209–215) directly contacts the N-terminal jellyroll domain of LptC (residues 60–69). Close-up view of the LptC–LptG interface, showing residues within 3.5 Å.

Native MS analysis revealed robust binding of both LPS types to LptB₂FG, with distinct adduct peaks corresponding to lipid-bound complexes (Supplementary Fig. S14). A clear preference was observed for *Pa*-LPS over *Ec*-LPS, as shown by higher signal intensity and multiple binding events at equivalent LPS concentrations. This affinity difference suggests that the transporter has adapted to recognise native *P. aeruginosa* LPS more tightly, likely due to subtle differences in the lipid A or core oligosaccharide structures between species (Supplementary Fig. S15).

Nevertheless, the transporter remains capable of recognising and interacting with *Ec*-LPS, as also evidenced by the challenges encountered in optimising the purification of the substrate-bound LptB₂FGC complex (Supplementary Fig. S1). Overall, these findings are consistent with prior reports showing that LptB₂FG(C) complexes purified from *E. coli* often contain endogenous *Ec*-LPS irrespective of the species of origin, underscoring the transporter's baseline promiscuity[28–31,34]. Our data refine this understanding by demonstrating that, while LptB₂FG can bind LPS from multiple sources, it exhibits measurable specificity that likely reflects physiological optimisation.

**Key determinants of LPS involved in binding to the LptB₂FG(C) complex**

To dissect the specific features of LPS required for recognition by the LptB₂FG(C) complex, we performed native MS experiments using defined LPS analogues: Re-LPS, lipid A, and 4′-monophosphoryl lipid A (MPLA) (Fig. 4a). These compounds differ progressively in their structural complexity: Re-LPS contains lipid A and two Kdo (3-deoxy-D-manno-oct-2-ulosonic acid) sugars; lipid A lacks the Kdo moieties; and MPLA is missing both Kdo and the phosphate group at position 1 of the glucosamine backbone. Only Re-LPS has been used previously to investigate Lpt system functionality[17,47,48]. Because of the lack of commercial availability of these variants from *P. aeruginosa*, we used *E. coli* versions for our study.

We incubated the LptB₂FG/LptB₂FGC solution with increasing concentrations of Re-LPS (0.1 μM to 5 μM) and recorded mass spectra under our optimised conditions. We could detect the presence of 2.2 kDa adduct in both LptB₂FG and LptB₂FGC charge states, consistent with Re-LPS (MW = 2239 Da) binding (Fig. 4b). The native mass spectra revealed a higher degree of total Re-LPS binding to LptB₂FGC than LptB₂FG (Fig. 4b, right inset). To quantify the binding affinity, we determined the apparent dissociation constant ($K_D$) for individual lipid-binding events to LptB₂FG and LptB₂FGC using a sequential lipid-binding model (Supplementary Fig. S16)[49,50]. In line with total Re-LPS binding quantification, LptB₂FGC displayed sub-micromolar binding affinity for Re-LPS ($K_{D(1)} = 0.65 \pm 0.02$ μM). By comparison, the apparent $K_{D(1)}$ of LptB₂FG:Re-LPS was more than twofold higher ($K_{D(1)} = 1.45 \pm 0.05$ μM) (Supplementary Fig. S16), suggesting that LptC contributes to high-affinity substrate binding. The increase in affinity may result from either a decreased off-rate of LPS from the central cavity or increased binding to periplasmic domains of the transporter. Because our data are equilibrium occupancies, we cannot distinguish between changes in association and dissociation rates here; accordingly, we present the reduced-off-rate explanation as a working hypothesis aligned with prior structural models of LptC-mediated gate opening[28–31]. While we do not infer periplasmic-domain engagement from our assays, we cannot completely exclude additional interactions under our conditions because the exact TM-helix position is unknown.

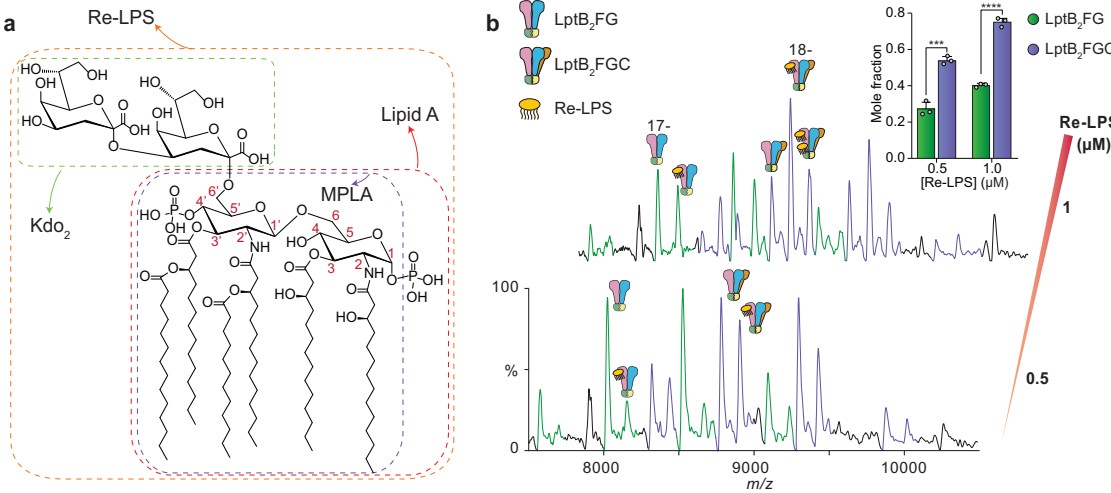

**Fig. 4 | Re-LPS binding to LptB2FG/LptB2FGC. a** Molecular structure of Re-LPS showing the two 3-deoxy-D-manno-oct−2-ulosonic acid (Kdo) along with the lipid A and MPLA substructures. **b** Native mass spectra of LptB$_2$FG/B$_2$FGC in the presence of increasing concentrations of Re-LPS. Charge state series indicating lipid binding were observed, and the total amount of lipid-bound protein complex was calculated. The bar charts show the quantification of Re-LPS-bound species and suggest that Re-LPS interacts more tightly with LptB$_2$FGC than LptB$_2$FG. Data are expressed as mean ± standard deviation (SD) of three biological replicates ($n$ = 3). Statistical significance was assessed using multiple $t$ tests (two-sided, unpaired) with Holm-–Šidák correction for multiple comparisons (*** $p$ = 0.00051; **** $p$ = 0.00002).

We then investigated the structure-binding relationships for protein-substrate interaction to elucidate the essential moieties for LPS binding. We incubated the LptB$_2$FG/LptB$_2$FGC solution with lipid A and 4′-monophosphoryl lipid A (MPLA) at increasing concentrations (Fig. 5a). Using native MS, we observed both lipids bound to LptB$_2$FGC at both concentrations employed. Quantification of the relative abundance of lipid-bound LptB$_2$FGC indicated a significant decrease of binding affinity for lipid A and MPLA, with the latter being the weakest interacting species (Fig. 5a). In the case of LptB$_2$FG, we again observed lower overall lipid binding with a similar trend regarding the relative affinities of the different species (Supplementary Fig. S17). These observations suggest that the Kdo units in the inner core of LPS are important moieties for the interaction between LptB$_2$FGC and its substrates. Similarly, the phosphate group on C1 likely engages in critical interactions with residues from LptB$_2$FGC TMDs. The decreased binding of lipid A and MPLA to LptB$_2$FG compared to LptB$_2$FGC also suggests that LptC is actively involved in recognising portions with the Kdo sugars and the C1-bound phosphate group.

Consistent with these binding trends, structural analyses and MD simulations (detailed below) indicate that the phosphate group on the glucosamine unit is positioned in the upper region of the LPS binding pocket and interacts with conserved positively charged residues, such as Lys33 in LptF. Comparisons with existing LptB$_2$FG(C)−LPS structures show species-dependent variation in this interaction, highlighting a flexible but essential recognition mechanism for LPS transport (Supplementary Table S2). Together, these data underscore the critical role of both the Kdo sugars and the C1 phosphate group in stabilising LPS binding within the transporter.

## Lipid binding to the LptB$_2$FG(C) complex

Recent studies indicated that cardiolipin (CDL) aids in MsbA-mediated LPS transport[51], so we sought to probe if there is a similar relationship between LptB$_2$FG/LptB$_2$FGC and CDL. In particular, we chose 14:0 CDL to be consistent with the 14 carbon lipid tails of Re-LPS, and incubated this with the LptB$_2$FG/LptB$_2$FGC complex, we could observe ~ 50% of LptB$_2$FGC bound to CDL at 2.5 μM concentration (Fig. 5b). Apparent $K_D$ calculation indicated a similar trend when comparing LptB$_2$FG and LptB$_2$FGC interaction with CDL to that of LPS (Supplementary Fig. S18). Indeed, CDL binds more tightly to LptB$_2$FGC ($K_{D(1)}$ = 3.07 ± 0.09 μM) than LptB$_2$FG ($K_{D(1)}$ = 4.80 ± 0.14 μM). This finding suggests a role for

LptC in interacting with this phospholipid. We set out to test whether CDL influences LPS binding to LptB$_2$FGC. To this end, we added increasing concentrations of CDL (1.0 to 5 μM) to a solution containing LptB$_2$FG(C):Re-LPS complex (Supplementary Fig. S19). Following CDL addition, native MS indicated that the total amount of Re-LPS bound to the protein decreased substantially in a concentration-dependent manner (Fig. 5b and Supplementary Fig. S19). These results suggest that the binding cavity of LptB$_2$FG is capable of recognising CDL.

From a physiological standpoint, these findings suggest that LptB$_2$FGC may be sensitive to local membrane lipid composition, particularly under conditions where cardiolipin is enriched, such as during stress responses, antibiotic treatment, or the stationary phase. This could reflect a regulatory mechanism by which cardiolipin modulates LPS extraction efficiency or prioritises substrate selection under specific conditions.

Our MD simulations (see below) point to shared contact sites with Re-LPS, including interactions with charged residues in TM1 and TM5 helices. Whether CDL acts purely as a competitive ligand or as a functional co-factor influencing transport dynamics is an open question, meriting further investigation.

## MD simulations highlight key coordinating residues for lipid binding in the central cavity

To explore the molecular basis for lipid recognition within the LptB$_2$FG binding cavity, we performed molecular dynamics (MD) simulations with either Re-LPS, Lipid A, MPLA or CDL bound to the central cavity. We then calculated the protein-lipid contacts made within 4 Å, over the course of three 500 ns simulations. Despite the asymmetry between LptF and LptG subunits, common domains were observed to interact with the bound lipids, with the majority of contacts formed with TM1, TM2 and TM5 helices (Fig. 6). For LptF, the bound lipids interacted with I25, I26, G29, R30 and K33 (TM1), L62, I63 and L66 (TM2), D154 and S246 in the periplasmic domain, and L305, I308, L309, M312, L315, A316 and I319 (TM5). For LptG contacts were made with I23, L26, A27, F30, I33, D34, N37 and D38 (TM1), R59, M63, M66 and I70 (TM2), R132 (TM3), K145, R146, E240 and R265 (periplasmic domain) and V308, F312, R315 and D319 (TM5). These residues likely contribute to a combination of hydrophobic and electrostatic interactions, with the arginine and lysine residues engaging with the negatively charged phosphate headgroups and the hydrophobic residues interacting with

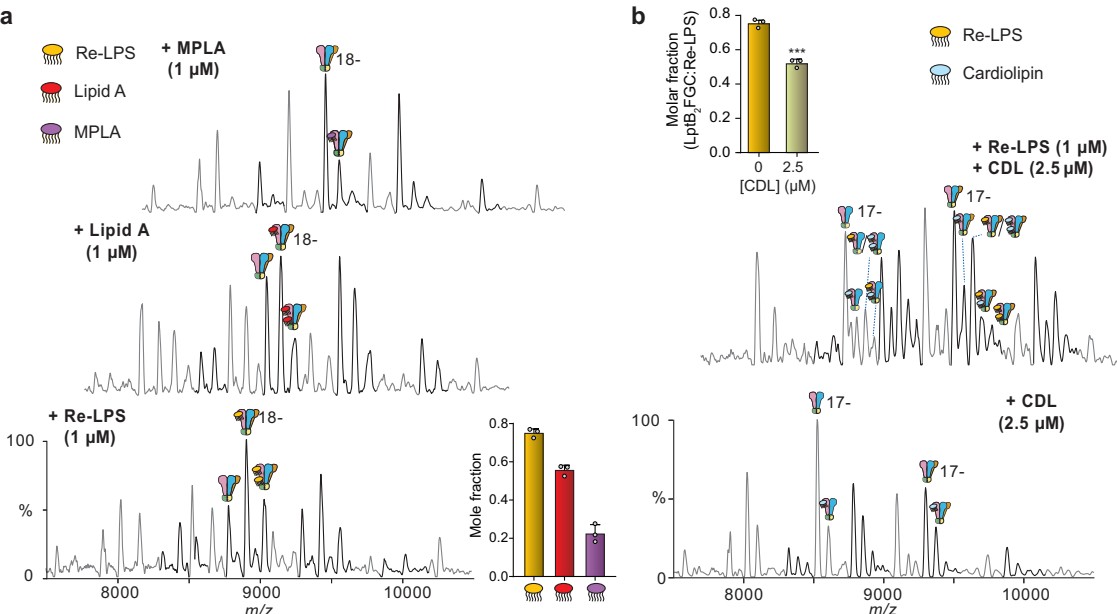

**Fig. 5 | Lipid binding to LptB₂FG/LptB₂FGC. a** Native mass spectra of LptB₂FG/ LptB₂FGC in the presence of Re-LPS, Lipid A, and MPLA. The quantification of lipid-bound species (bar charts) suggests that the amount of lipid binding decreases as the number of phosphate groups decreases (inset). Data are expressed as mean ± standard deviation (SD) of three biological replicates (n = 3). **b** Native mass spectra of LptB₂FG/LptB₂FGC in the presence of CDL and in the presence of CDL and LPS. The inset panel shows a quantification of the mole fraction of total Re-LPS-bound LptB₂FGC as a function of total [CDL]. Data are expressed as mean ± standard deviation (SD) of three biological replicates (n = 3). Statistical significance was assessed using the Welch's *t* tests (two-sided, unpaired; ***p = 0.00034).

the acyl chains of LPS and CDL. Notably, residues such as I25, K33, and L62 of LptF, and L26, D34, R132, and R317 of LptG correspond to key contact points identified in the *S. flexneri, E. coli, A. baylyi* and *K. pneumoniae* structures (Supplementary Figs. S20 and S21). Among the 38 cavity-contact residues highlighted by MD (17 LptF; 21 LptG), prior functional data exist for a subset of positions in enterobacteria (although not directly tested in *P. aeruginosa*), most notably hydrophobic/charged residues lining the cavity whose mutation compromises Lpt function or growth (e.g., LptF I25/R30/L62; LptG L26/R59/ I70/R132/V308/F312). These studies collectively support the idea that the central cavity residues we observe contacting acyl chains in MD are functionally important[28,29,33,34], although numbering and exact identities vary by species/construct.

Although the contact profiles for Re-LPS, Lipid A, MPLA, and CDL overlapped substantially, Re-LPS formed more persistent and extensive interactions, suggesting greater shape and charge complementarity within the binding cavity. In summary, these simulations provide mechanistic insight that complements our experimental data, identifying specific amino acids likely responsible for lipid selectivity.

## Discussion

Our combined structural, biochemical, and computational analyses redefine the apo state of the LptB₂FG(C) transporter and provide new insights into how this complex recognises and selects lipid substrates. Notably, we discovered that excess LPS co-purifies with the LptB₂FG(C) complex. By expressing the transporter in ClearColi cells, which produce tetra-acylated lipid IVA instead of typical hexa-acylated enterobacterial lipid A, we were able to deplete all lipids, enabling the production of a lipid-free LptB₂FG/LptB₂FGC complex. Subsequent cryo-EM analysis reveals three major conformations for LptB₂FG and two for LptB₂FGC, representing the substrate-free state of the transporter. Comparing our nanodisc-embedded LptB₂FG-II structure with the previously reported X-ray structure of LptB₂FG (PDB ID: 5X5Y)[35] revealed significant conformational changes in the transmembrane helices of the transmembrane domain, with a root mean square deviation (RMSD) of 2.36 Å over 846 Cα atoms (Supplementary

Fig. S9a). Importantly, our cryo-EM structures did not reveal any densities for LPS within the outward-facing cavity of TMDs. The LPS binding cavity in our structure has approximately half the volume observed in the X-ray structure, which has been considered the apo state structure so far in the field. This is also in contrast to all the existing cryo-EM structures of LptB₂FG and LptB₂FGC determined so far. The distinctive structural attributes characterising our cryo-EM LptB₂FG structures strongly suggest that we have captured this transporter's resting state or ligand-unbound conformation.

In comparison with the *V. cholerae* LptB₂FGC structure (6MJP), we observed substantial differences, indicating that the LptB₂FGC-I structure likely represents a distinct functional state (Supplementary Fig. S13d). Notably, our structure exposes Interface 1 directly to the membrane, while in *V. cholerae* LptB₂FGC, the transmembrane region of LptC obstructs this interface (see Supplementary Fig. S13). Conversely, our structures demonstrate a closed interface 2 (Supplementary Fig. S13), unlike *V. cholerae* LptB₂FGC, which displays an open Interface 2. It appears that the incorporation of the TM helix of LptC into Interface 1 pushes the rigid group 2 of LptF to rotate counterclockwise toward LptG (as observed from the top of the transmembrane domain), causing LptF to move further away from LptG (Supplementary Fig. S13d). Conversely, in *V. cholerae* LptB₂FGC, the rigid group 1 of LptF rotates clockwise, resulting in the opening of interface 2. The rigid body rotations of both group 1 and group 2 significantly increase the volume of the TMD binding cavity. This intriguing difference prompts us to reconsider whether *V. cholerae* LptB₂FGC represents a state where LptB₂FGC binds lipids containing both LPS and non-LPS constituents. The relevance of β-jellyroll movement may be crucial for the transportation of LPS across the periplasmic region. It is known that once expelled from the transporter, the LPS molecule rotates by about 90° to become localised within β-jellyroll domains of the bridge[52]. We believe the observed conformational changes reflect a degree of structural plasticity in the periplasmic components that may coordinate with ATPase cycling and LPS handoff.

It is interesting to note that, although LPS-bound, the recent *A. baylyi* LptB₂FG structure (8FRM) adopts a closed conformation at

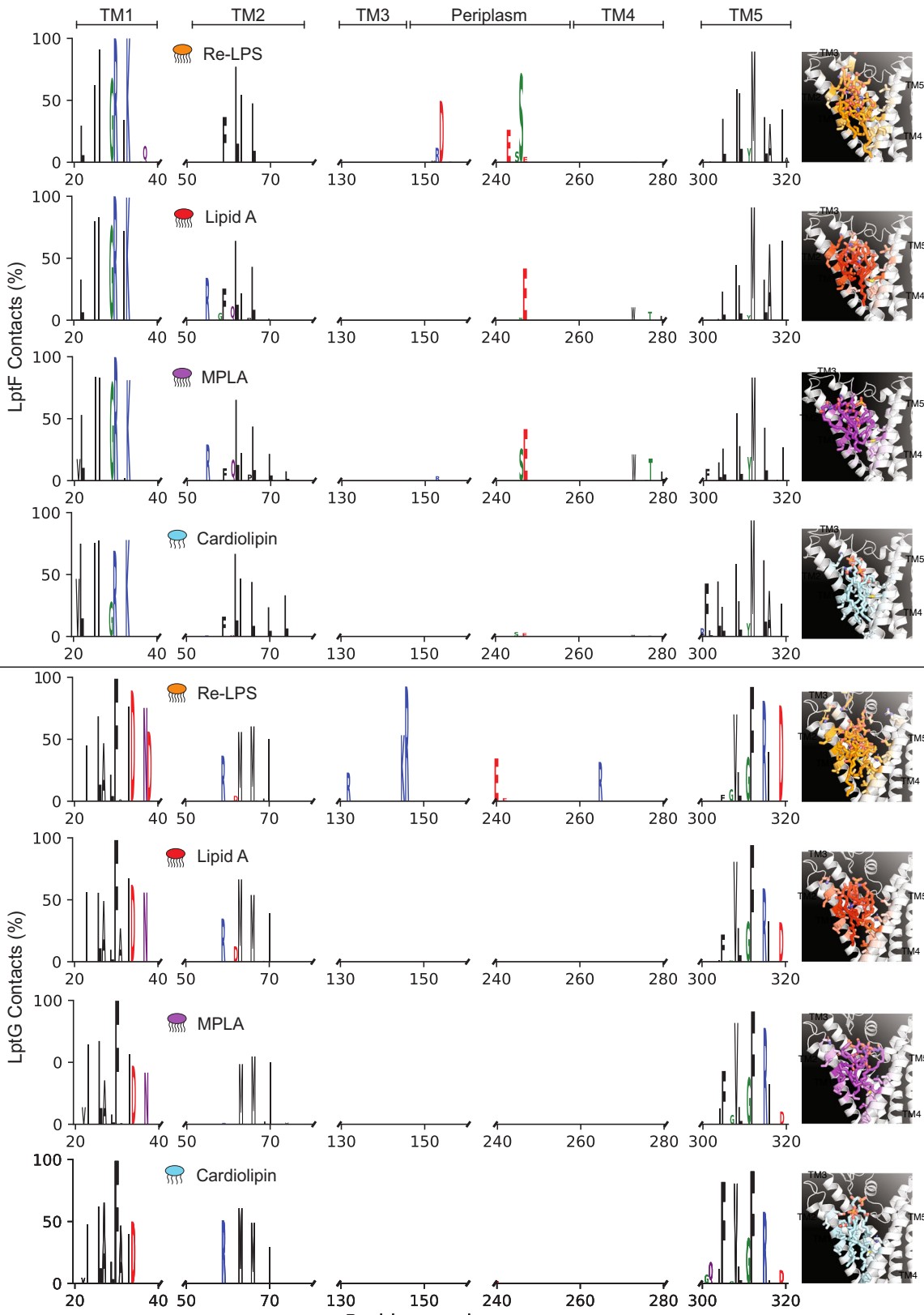

**Fig. 6 | Lipid binding to LptB₂FG from MD simulations.** Logoplots are shown for lipid interactions between Re-LPS, Lipid A, MPLA and cardiolipin and both LptF and LptG. The contacts were calculated with respect to the percentage of the simulation time a given residue remained within 4 Å of the bound lipid. Snapshots are shown of both LptF and LptG with the bound lipid, with the interacting residues coloured on a white-to-coloured scale for the percentage of contacts made.

Interface 2, which is similar to our substrate-free *P. aeruginosa* structure. The structural analysis suggests that this *A. baylyi* conformation captures a state in which LPS has entered the cavity through interface I.

Importantly, our work provides a structural basis for the known genetic incompatibility between *P. aeruginosa* and *E. coli* LptC. Our cryo-EM structures of *P. aeruginosa* LptB$_2$FGC revealed a unique heterotrimeric interface between LptC, LptF, and LptG, with species-specific β-jellyroll contacts that likely underlie the failure of *P. aeruginosa* LptC to complement the *E. coli* system[32]. This finding expands our understanding of Lpt system diversity and suggests that LptC–LptF/G interactions evolve in a coordinated, species-adapted manner.

Our lipid-binding studies further revealed that, beyond LPS, the transporter can accommodate cardiolipin (CDL), a lipid previously implicated in MsbA-mediated LPS transport[51]. The ability of CDL to bind and displace LPS (with a model LPS) in a concentration-dependent manner, combined with MD simulations showing overlapping contact residues, raises the possibility that non-LPS lipids such as CDL might transiently interact with or regulate the transporter under specific cellular conditions (e.g., stress, membrane remodelling)[53–58]. This proof-of-concept observation adds a layer of physiological relevance to the transporter's substrate selectivity and suggests that LptB$_2$FGC may function within a lipid-rich regulatory environment, which remains to be tested in future studies. The fact that our native MS analysis indicated that the transporter is not selective to LPS and can recognise CDL, and previous *Sf*LptB$_2$FGC, where lipid-like acyl chains were present at Interface 2[25], we hypothesise that the accessible interface 2 in *V. cholerae* LptB$_2$FGC may serve as a plausible route for the egress of these non-LPS lipids, which awaits confirmation in the future.

In summary, our work highlights the conformational flexibility within the Lpt system and provides a molecular basis for LPS recognition and transport. Due to the critical importance of LPS to bacterial survival, impeding its biogenesis constitutes a successful approach to antimicrobial drug discovery. To this end, different studies uncovered peptide and small molecule-based inhibitors of LPS transport by disrupting the protein-protein interactions in the periplasmic bridge formed by LptC, LptH, and LptD[59–62]. Moreover, recent reports suggest that targeting the IM complex LptB$_2$FG is possible, paving the way for the development of a new class of antibacterial compounds[30,63]. The LPS-free structures we report offer a valuable template for future rational design of small molecules that could lock the transporter in its inactive conformation or compete with lipid substrates at defined entry interfaces.

## Methods

### LptB$_2$FGC expression and membranes isolation
The plasmids used for overexpression of *P. aeruginosa* LptB$_2$FG and LptB$_2$FGC were a kind gift from Yihua Huang (Chinese Academy of Sciences, Beijing, China). The plasmids were constructed as previously described (Luo et al., 2017) and consisted of pQLink-LptBFG or pQLink-LptBFGC, whereby each gene (*lptB*, *lptC*, *lptF* and *lptG*) had been amplified from *P. aeruginosa* PAO1-LAC genomic DNA (47085D-5 ATCC) using PCR and inserted in the pQlink plasmid by ligation-independent pathway. To allow affinity purification, a 6 x His-tag was attached to the C-terminus of LptB. All plasmids were amplified by transforming them into *E. coli* Stellar Competent Cells (Takara), and the DNA sequences were verified by Sanger sequencing.

Initial expression and purification were performed adapting previously described protocol[33]. The amplified plasmid was transformed in *E. coli* BL21(DE3) (Lucigen). Colonies were inoculated into 100 ml LB media and grown overnight at 37 °C supplemented with 100 µg/ml ampicillin. One litre of LB in 2 litre shaker flasks was inoculated with 10 ml of overnight culture and grown at 37 °C to a density of 0.6-0.7 at 600 nm at 37 °C, and the expression of the recombinant protein complex was induced by 0.5 mM isopropyl-β-D-thiogalactopyranoside (IPTG) at 20 °C for 16 h. Cells were collected by centrifugation at

5000 x *g* for 10 min at 4 °C. Cell pellets were resuspended in buffer containing 150 mM NaCl, 20 mM Tris-HCl (pH 8.0) and stored at − 80 °C.

Resuspended cells were thawed and supplemented with EDTA-free protease inhibitor cocktail (Roche) and 10 µg/ml DNaseI. The cell suspension was passed several times through an M-110 PS microfluidiser (Microfluidics) at 15,000 psi. Insoluble material was pelleted by centrifugation at 20,000 x *g* for 20 min at 4 °C. The supernatant was and then ultracentrifuged at 200,000 x *g* for 1 h to collect membrane fractions. Membranes were resuspended in ice-cold buffer containing 250 mM NaCl, 20 mM Tris-HCl (pH 8.0) and 20% glycerol. Resuspended membranes were used either directly or flash frozen in liquid nitrogen and stored at − 80 °C.

To fully remove LPS adducts, we then expressed LptB$_2$FG and LptB$_2$FGC in ClearColi BL21(DE3) (Lucigen), an *E. coli* strain holding different gene knockouts that stop the biosynthetic pathway of LPS to a precursor, lipid IVA. The amplified pQLink-LptBFGC plasmid was transformed in ClearColi BL21(DE3) through electroporation, and the plates were incubated for 32-40 h at 37 °C. Colonies were inoculated into 100 ml LB media and grown overnight at 37 °C supplemented with 100 µg/ml ampicillin. One litre of LB in 2 litre shaker flasks was inoculated with 40 ml of overnight culture and grown at 37 °C to a density of 0.6-0.7 at 600 nm at 37 °C, and the expression of the recombinant protein complex was induced by 0.5 mM IPTG at 20 °C for 16 h. The steps for cell lysis and membrane isolation are the same as the ones described above.

### LptB$_2$FG/LptB$_2$FGC purification
The protein was solubilised from the membrane fraction with 250 mM NaCl, 20 mM Tris-HCl (pH 8.0), 20% glycerol, 2% (w/v) DDM, 2% (w/v) OGNG (Anatrace) for 16 h at 4 °C. Insoluble material was removed by centrifugation at 20,000 x *g* for 20 min at 4 °C. The supernatant was filtered before loading onto a 5 ml HisTrap-HP column (GE Healthcare, Piscataway, NJ) equilibrated in 150 mM NaCl, 20 mM Tris-HCl (pH 8.0), 20 mM imidazole, 1% (w/v) DDM, and 1% (w/v) OGNG. After the clarified supernatant was loaded, the column was initially washed with 500 ml (100 CVs) of 150 mM NaCl, 20 mM Tris-HCl (pH 8.0), 20 mM imidazole, 1% (w/v) DDM, and 1% (w/v) OGNG, and washed again with 50 ml of 150 mM NaCl, 20 mM Tris-HCl (pH 8.0), 80 mM imidazole and 0.1% DDM. The bound protein was eluted with 150 mM NaCl, 20 mM Tris-HCl (pH 8.0), 300 mM imidazole and 0.03% DDM.

The protein was concentrated to 2.5 ml an Amicon Ultra-15 concentrator unit (Millipore) with a molecular cut-off of 100 kDa and buffer exchanged to 150 mM NaCl, 20 mM Tris-HCl (pH 8.0), 10% glycerol and 0.03% DDM using a PD-10 desalting column. The sample was further concentrated and loaded onto the Superdex 200 size exclusion chromatography (SEC) column in 150 mM NaCl, 20 mM Tris-HCl (pH 8.0), 10% glycerol and 0.03% DDM. Protein concentration was measured using a DS-11 FX Spectrophotometer (DeNovix).

### Lipids, nucleotides, and peptide preparation
Unless stated otherwise, all lipids [Kdo2-Lipid A (Re-LPS), 1,2-dimyristoyl-sn-glycero-3-phospho-(1′-rac-glycerol) (14:0 PG, DMPG), 1,2-dimyristoyl-sn-glycero-3-phosphoethanolamine (14:0 PE, DMPE) 1′,3′-bis[1,2-dimyristoyl-sn-glycero-3-phospho]-glycerol (14:0 cardiolipin, CDL) used in this study were obtained from Avanti Polar Lipids powders and stock solutions were prepared following a previously established method[38]. Full-length *E. coli* LPS and *P. aeruginosa* LPS are obtained from Sigma (catalogue numbers L3012 and L8643, respectively).

### Native MS experiments
Purified LptB$_2$FG(C) was buffer exchanged into MS Buffer (200 mM ammonium acetate + 0.04% (w/v) [G1]-OGD) using a centrifugal buffer exchange device (Micro Bio-Spin 6, Bio-Rad) as previously described. The freshly buffer-exchanged protein was kept on ice, with protein

concentration measured as before. The protein samples were diluted as desired in 200 mM ammonium acetate buffer with detergent as necessary and loaded into a gold-coated capillary Clark borosilicate capillary (Harvard Apparatus) prepared in the laboratory. The experiments were performed using a Q Exactive UHMR Hybrid Quadrupole-Orbitrap mass spectrometer (Thermo Fisher Scientific). Typically, 2 µl of buffer-exchanged protein solution was electrosprayed in negative polarity from in-house prepared gold-coated borosilicate capillaries. The instrument parameters used for data collection were: capillary voltage 1.1 kV, S-lens RF 100%, quadrupole selection from 1000 to 20,000 m/z range, HCD collision energy 200–300 V, source fragmentation 0 V, in-source trapping 200–300 V in negative mode. The ion transfer optics was set as follows: injection flatapole − 5 V, inter-flatapole lens − 4 V, bent flatapole − 2 V, transfer multipole 0 V. The resolution of the instrument was 8750 at m/z = 200 (transient time of 64 ms), argon pressure in the HCD cell was maintained at approximately at $8 \times 10^{-10}$ mbar and source temperature was kept at 200 °C. The noise level was set at 3 rather than the default value of 4.64. Calibration of the instruments was performed using 10 mg/ml solution of caesium iodide in water. Where required, baseline subtraction was performed to achieve a better-quality mass spectrum. Where required, baseline subtraction was performed to achieve a better-quality mass spectrum. Data were analysed using the Xcalibur 3.0 (Thermo Scientific), NaViA[64], and UniDec (www.unidec.chem.ox.ac.uk)[65] software packages.

Lipid and nucleotide binding experiments were performed by diluting them in 200 mM ammonium acetate supplemented with 0.04% [G1]-OGD. Native MS experiments were performed at a total LptB$_2$FG(C) protein concentration of 3 µM.

Peak intensities were extracted using UniDec, and the ratios of the intensity of the ligand bound peak versus the total intensity of all observed species were calculated. The apparent dissociation constant ($K_D$) values for individual lipid-binding events to LptB$_2$FG and LptB$_2$FGC were calculated through a sequential lipid-binding model using a previously described Python script[50]. When necessary, a Welch's T-test was used to evaluate statistically significant differences and data were plotted using GraphPad Prism 8.0. Error bars in the plots indicate standard deviation, and all experiments were performed in triplicate ($n = 3$).

## ATPase activity assay

The ATPase activity of LptB$_2$FG, LptB$_2$FG(C), and LptB$_2$FG(C) in native MS buffer were monitored using the ATPase Assay Kit from Abcam (ab270551) following the manufacturer's protocol. To monitor the ATPase activity, all assay components were allowed to warm up to room temperature prior to use, including the protein. The assay buffer consisted of SEC purification buffer or native MS buffer supplemented with 1 mM ATP and 5 mM Mg$^{2+}$. Solutions were added to each well of a 96-well plate (Greiner 96F-Bottom) in the following order: (i) assay buffer (100 µL), (ii) protein (100 µL of a 0.5 µM solution in the relevant buffer), (iii) wait 30 min, (iv) PiColorLock$^{TM}$ (50 µL), (v) wait 2 min, (vi) stabiliser (20 µL). Half an hour after adding the stabiliser, the absorbance values at 610 nm (A$_{610}$) were determined using a PerkinElmer EnSight multimode plate reader. To determine the molar amount of phosphate generated by each protein preparation, a standard curve was determined according to the instructions given in the protocol provided with the kit. ATPase activities of all samples were determined using the mean value of the samples according to the linear regression of standards. The data were plotted using GraphPad Prism 8.0. Error bars in the plots indicate standard deviation, and all experiments were performed six times ($n = 6$).

## Nanodisc preparation

To assemble LptB$_2$FG(C) into nanodiscs, a mixture containing 10 µM LptB$_2$FG(C), 30 µM membrane scaffold protein (MSP; 1D1), and 300 µM E. coli total extract lipid was incubated for 15 min at room temperature. After, 0.8 mg/ml prewashed Bio-Beads (Bio-Rad) was added. The resultant mixture was incubated for 1 h on ice, followed by overnight incubation at 4 °C. The protein-nanodisc solution was filtered through 0.22 µm nitrocellulose filter tubes to remove the Bio-Beads. To separate free nanodiscs from LptB$_2$FG(C)-loaded nanodiscs, the filtered protein-nanodisc solution was purified using a Superdex 200 column (GE Healthcare) equilibrated with 20 mM Tris-HCl (pH 7.5) and 100 mM NaCl. Fractions corresponding to the size of the LptB$_2$FG(C)-nanodisc complex were collected for cryo-EM.

## Electron microscopy sample preparation

The LptB$_2$FG(C)-nanodisc sample was concentrated to 1 mg/ml. A 2.5 µl sample was applied to glow-discharged holey carbon grids (Quantifoil Cu R1.2/1.3, 300 mesh), blotted for 5 s, and then plunge-frozen in liquid ethane using a Vitrobot (Thermo Fisher). The grids were transferred into cartridges. For high-resolution data collection, the sample grids were loaded into a Titan Krios cryo-electron microscope operated at 300 kV equipped with a Gatan BioQuantum imaging filter (GIF) and a K3 summit direct electron detector (Gatan). The micrographs were recorded using Latitude software (Gatan) with counting mode at nominal × 81,000 magnification corresponding to a calibrated pixel size of 1.08 Å (super-resolution, 0.54 Å/pixel) and a defocus range of − 1 to − 2.5 µm. To remove inelastically scattered electrons, the slit width was set to 20 eV. Each micrograph was exposed for 2 s with a total specimen dose of 50 e-/Å2, and 40 frames were captured per specimen area.

## Cryo-EM data processing

The LptB$_2$FG(C) dataset was processed using a BaR[45] protocol. Super-resolution image stacks were aligned and binned using patch motion correction (cryoSPARC)[66] with a binning factor of 2 to give a final pixel size of 1.08 Å /pixel. Contrast transfer functions (CTFs) were estimated using the patch CTF in cryoSPARC. After manual inspection to discard poor images and to estimate particle size, the blob picker in cryoS-PARC was used to select particles from subsets of micrographs. These particles were classified in several rounds of 2D classification, and clear templates were selected for template picking in cryoSPARC. The template picker was used to select initial particle sets. Several iterative rounds of 2D classification were used to clean these particle sets with different circular masks to account for different particle sizes. Featureless classes were removed from each step to obtain cleaned heterogeneous particle stacks for further processing. The 2D class averages containing clear 2D features were selected and used to generate a Topaz training model. The trained model was then used to pick particles using Topaz extract in cryoSPARC.

From heterogeneous particle sets, particles were classified, and final maps were solved with the use of BaR. In brief, ab initio methods were used to build initial 3D maps from the selected classified particles, and particles were then retrieved based on the maps. To build the initial maps, particles were separated using 2D classification paired with 3D ab initio and heterogeneous classifications. These initial classes were used to retrieve particles from the topaz-picked particle stack. 3D heterogeneous refinement using the ab initio classes, determined from the build phase of BaR, was applied to the cleaned heterogeneous particle sets. The new particle subsets were then cleaned using multiple rounds of 2D and 3D ab initio classifications. Non-uniform refinement using cryoSPARC was used to refine 3D reconstructions. A soft masks that covers the LptB$_2$FG or LptB$_2$FGC were used for local refinement. The resulting maps corresponding to the Lpt transporters exhibited high quality, allowing for the construction of models.

## Model building and refinement

The models for LptB$_2$FG-I, LptB$_2$FG-II, LptB$_2$FG-III, LptB$_2$FGC-I, and LptB$_2$FGC-II were constructed based on cryo-EM maps with resolutions of 3.34 Å, 3.26 Å, 3.31 Å, 3.26 Å, and 3.61 Å, respectively. Initial structures of LptB$_2$FG, LptB$_2$FGC were generated using Alphafold 2[67] (using uniport accession numbers Q9HVV6, Q9HXH5, Q9HXH4 and Q9HVV8) and fitted into the density maps as starting models using Chimera[68]. Subsequent model rebuilding was carried out using Coot[69], and structural refinements were performed using phenix.real_space_refine from the PHENIX suite[70]. The final atomic model was evaluated using MolProbity[71]. Detailed statistics regarding data collection, 3D reconstruction, and model refinement can be found in Supplementary Table S1.

## Molecular Simulation setup

All simulations were run using GROMACS 2021[72]. The LptBFG complexes were positioned in a membrane using the Martini 3 coarse-grain (CG) force field and solvated with water and 0.15 M NaCl to neutralise the system[73]. The membranes were constructed using *insane* with a 7:2:1 ratio of PE:PG:CDL lipids, using the MemProtMD pipeline[74,75]. An elastic network of 1000 kJ mol$^{-1}$ nm$^{-2}$ was applied between all backbone beads between 0.5 and 1 nm. Electrostatics were described using the reaction field method, with a cut-off of 1.1 nm using the potential shift modifier and the van der Waals interactions were shifted between 0.9-1.1 nm. The systems were first energy minimised by the steepest descent algorithm to 1000 kJ mol$^{-1}$ nm$^{-1}$ and then simulated for a total of 1 μs. The temperature and pressure were kept constant throughout the simulation at 310 K and 1 bar, respectively, with protein, lipids and water/ions coupled individually to a temperature bath by the V-rescale method[76] and a semi-isotropic Parrinello-Rahman barostat[77]. The final snapshots from the CG simulations were then converted back to an atomistic description using CG2AT[78]. LPS and CDL coordinates were positioned into the central cavity of LptBFG based on the coordinates from PDB ID 6S8H[29], using PyMOL. Converted all-atom simulations of the LptBFG complexes were performed without position restraints for a total of 500 ns, and run in triplicate. In all cases, a 2 fs timestep was used, in an NPT ensemble with V-rescale temperature coupling at 310 K[76] and a semi-isotropic Parrinello-Rahman barostat at 1 bar, with protein, water/ions and, if included, lipids coupled individually[77]. Electrostatics were described using PME, with a cut-off of 1.2 nm and the van der Waals interactions were shifted between 1-1.2 nm. The tip3p water model was used, the water bond angles and distances were constrained by SETTLE[79]. H-bonds were constrained using the LINCS algorithm[80]. Analysis was performed using MDAnalysis[81] and visualised in PyMOL (Schrödinger, LLC. 2015).

## Generation of plasmids and complementation assays

The DNA fragments for cloning were amplified by PCR using the genomic DNA of *P. aeruginosa* PAO1 or the plasmid pQLinkN-yh-LptB-KL-6×His-FGC as the template. All constructs generated in this work were verified by DNA sequencing.

The plasmid constructs to express LptB protein variants were generated by directionally cloning the coding sequence of each gene of interest or its recombinant variant into the shuttle vector pME6032[82], downstream of the IPTG-inducible P*tac* promoter. The pME6032 derivatives were introduced into *P. aeruginosa* by transformation using chemically competent cells.

Complementation assays were performed by pre-culturing the strains of interest in Mueller-Hinton broth supplemented with 0.01% rhamnose until the late-exponential phase. Cells were collected by centrifugation and resuspended in saline at an OD$_{600}$ = 1, and 5 μL aliquots of serial ten-fold dilutions in saline were plated onto Mueller-Hinton agar plates supplemented or not with 0.0.1% rhamnose or 0.5 mM IPTG. Images were taken after 16 h of incubation at 37 °C.

## Sequence alignment

Sequence alignments were performed using the NPS@ClustalW[83,84] web service and rendered using the ESPript 3.0 web service[85].

## Reporting summary

Further information on research design is available in the Nature Portfolio Reporting Summary linked to this article.

## Data availability

Cryo-EM maps and coordinates have been deposited in the EMDB and PDB, respectively, with accession numbers EMD-47084 and PDB 9DOH (LptB$_2$FG-I); and PDB 9DOK (LptB$_2$FG-II); EMD-47086 and PDB 9DOO (LptB$_2$FG-III); EMD-47088 and PDB 9DOQ (LptB$_2$FGC-I) and EMD-47089 and PDB 9DOR (LptB$_2$FGC-II). The mass spectrometry raw data have been deposited to the ProteomeXchange Consortium via the PRIDE[86] partner repository with the dataset identifier PXD069660. MD simulation parameters, input, and output files have been deposited in Zenodo (https://zenodo.org/records/17408980). Previous structures used for comparison are 5X5Y, 6MHU, 6S8H, 5L75, 7EFO, 8FRM, 6MJP, 6MIT, 6MI7, 8FRP and 6S8N. Source data are provided in this paper.

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

## Acknowledgements

We gratefully acknowledge funding from the Medical Research Council (MR/V028839/1, C.V.R.), the Royal Society (URF\R1\211567, J.R.B.), the UKRI Frontier Research Guarantee (EP/Y036158/1, J.R.B.), the American Heart Association (25CDA1455062, C-C.S.), the Italian Ministry of University and Research (MUR) with the grants Excellence Departments (art. 1, commi 314-337 Legge 232/2016) to the Department of Science of the University Roma Tre and PRIN 2020 (20208LLXEJ, F.I.) and Sapienza University of Rome (Grandi Attrezzature Scientifiche 2021 n. GA12117A8711CC9F - INCENTIVE MS, A.M.). P.J.S. lab was funded by Wellcome (208361/Z/17/Z), MRC (MR/Z504245/1), EPSRC, BBSRC (BB/P01948X/1, BB/Y007603/1, BB/Y003187/1, BB/Y003306/1) and the Howard Dalton Centre. P.J.S. acknowledges Sulis at HPC Midlands +, which was funded by the EPSRC (EP/T022108/1) and the University of Warwick Scientific Computing Research Technology Platform for computational access. This research was, in part, supported by the National Cancer Institute's National Cryo-EM Facility at the Frederick National Laboratory for Cancer Research under contract 75N91019D00024 and by the Clinical and Translational Science Collaborative of Cleveland, UM1TR004528 from the National Centre for Advancing Translational Sciences (NCATS) component of the National Institutes of Health and NIH roadmap for Medical Research. J.B.S. was supported by a BBSRC Doctoral Training Partnership studentship. R.A.C. was funded by Wellcome (208361/Z/17/Z). A.T. was supported by the American Society for Pharmacology and Experimental Therapeutics. The plasmids used for overexpression of P. aeruginosa LptB$_2$FG and LptB$_2$FGC were a kind gift from Yihua Huang (Chinese Academy of Sciences, Beijing, China).

## Author contributions

Conceptualisation of project: F.F., C.V.R. and J.R.B.; Methodology: F.F., M.C., Y.W., L.H.U., J.B.S., R.A.C., D.R., A.M., P.J.S., F.I., C-C.S., C.V.R. and J.R.B.; Investigation: F.F., M.C., Y.W., L.H.U., J.B.S., A.T., R.A.C., P.J.S., F.I., C-C.S. and J.R.B.; Funding acquisition: F.F., A.M., P.J.S., F.I., E.W.Y., C.V.R. and J.R.B.; Project administration: P.J.S., F.I., E.W.Y., C-C.S., C.V.R. and J.R.B.; Supervision: A.M., P.J.S., F.I., C.V.R. and J.R.B.; Writing—original draft: F.F., C-C.S. and J.R.B.; Writing—review and editing: F.F., F.I., E.W.Y., C-C.S., C.V.R. and J.R.B. All authors commented on the final version of the manuscript.

## Competing interests

C.V.R. is the founder and consultant of OMass Therapeutics. All other authors have no competing interests.

## Additional information

[1]Department of Biochemical Sciences, Sapienza University of Rome, Rome, Italy. [2]Department of Chemistry, University of Oxford, Oxford, UK. [3]Department of Science, Roma Tre University, Rome, Italy. [4]Department of Biology, University of Oxford, Oxford, UK. [5]Department of Chemistry and Chemical Biology, TU Dortmund University, Dortmund, Germany. [6]Department of Biochemistry, University of Oxford, Oxford, UK. [7]Department of Pharmacology, Case Western Reserve University School of Medicine, Cleveland, OH, USA. [8]School of Physiology, Pharmacology & Neuroscience, University Walk, Bristol, UK. [9]Department of Drug Chemistry and Technologies, Sapienza University of Rome, Rome, Italy. [10]School of Life Sciences, Gibbet Hill Campus, The University of Warwick, Coventry, UK. [11]Kavli Institute for Nanoscience Discovery, University of Oxford, Dorothy Crowfoot Hodgkin Building, University of Oxford, Oxford, UK. ✉e-mail: cxs670@case.edu; carol.robinson@chem.ox.ac.uk; jani.bolla@biology.ox.ac.uk

