## [Peer Review File · Nature Communications]

Structural snapshots of *Pseudomonas aeruginosa* LptB2FG and LptB2FGC reveal insights into lipopolysaccharide recognition and transport

Corresponding Author: Dr Jani Bolla

Version 0:

Reviewer comments:

Reviewer #1

(Remarks to the Author)

The authors have provided a satisfactory rebuttal to my prior criticisms. I'm grateful to see that my comments proved to be useful for the authors in providing a much more interesting revised manuscript. I should mention that the statement in lines 287-288 is redundant considering that the same point was made in lines 282-283. Finally, regarding my query in Reviewer 1 point 4, the authors reply states "In theory, the phosphate-to-phosphate distance (~10 Å for cardiolipin versus ~6 Å for lipid IVA) can be distinguished in cryo-EM when local resolution is better than 3 Å and the lipid is well ordered." However, I ask the authors to consider that their numbers can't be correct as the phosphate groups in cardiolipin are closer together, while those of lipid A are farther apart. The reference I was citing is extended data Figure 9 in Clairfeuille T, et al. Structure of the essential inner membrane lipopolysaccharide-PbgA complex. *Nature*. (2020) 584: 479-483. This paper resolved some previous errors that had been made by structural biologists who failed to adequately distinguish between cardiolipin and lipid A. Since the present authors are making the same distinctions, they should be clear about what it is that they are comparing. The relevant link is:

<https://www.nature.com/articles/s41586-020-2597-x/figures/13>

Reviewer #2

(Remarks to the Author)

The authors provided a very thorough response, and I have no further significant edits to suggest. I appreciate the detailed description of what manuscript edits were made in response to each point raised.

Reviewer #3

(Remarks to the Author)

The authors have improved the manuscript. I appreciate the work the authors put into revising the manuscript, especially by adding complementation studies, including data using native *Pseudomonas* LPS, and clarifying some important information in the text. Although the manuscript has been improved, there are still significant issues to address to justify several of the authors' conclusions.

Main issues:

1) The revised manuscript now shows that the tagged LptB protein complements in *Pseudomonas*. Specifically, the authors tested if relevant LptB proteins encoded in plasmids support growth of an lptB conditional strain in the absence of inducer. While this is a significant improvement of the study, the function should be more thoroughly tested for the following reasons. Previous studies have shown that although *E. coli* LptB-His has ATPase activity (in vitro) and complements the deletion of the lptB gene, it is partially functional in cells. It is functional enough to maintain growth but leads to significant defects in LPS transport, as demonstrated by the fact that cells with LptB-His are permeable (sensitive) to antibiotics and lptB-His can have either positive or negative effects in combination with specific lpt mutations. Whether the same is true or not in *Pseudomonas* remains unknown, but Fig. S3 shows that even though the depletion strain expressing LptB-KL-6xHis (used in this study) can grow without inducer, it grows significantly smaller colonies than a strain producing a His-less LptB-KL. They also show that a depletion strain producing LptB-6xHis grows better than LptB-KL-6xHis but colonies are still smaller

than colonies producing LptB. These data suggest that LptB-KL-6×His protein used in this study is partially functional, likely with significant defects in LPS transport based on the growth data. To describe it as functional is not appropriate. The effect of LptB-KL-6×His on the *Pseudomonas* Lpt transporter function remains uncharacterized. Whether structures would be different with a fully functional LptB is unknown, but it is a concern since the ATPase controls movement of the ABC transporter.

2) The authors now report that *Pseudomonas* LPS binds better to the *Pseudomonas* Lpt complex than the *E. coli* LPS (Fig S12). Unfortunately, they did not provide quantitative data. I could not find information about where the *Pseudomonas* LPS came from or how they purified it (unless I missed it, please add), but if the lack of quantitative data results from the fact that the *Pseudomonas* LPS is a mixture, I do not understand why they did not then purify it from any of the LPS mutants reported in the literature that would yield a more homogenous preparation. The lack of quantification is especially relevant to the issue about the biological significance of cardiolipin binding, which remains unclear mainly because there is no evidence that the authors' findings are relevant in *Pseudomonas* cells. Would cardiolipin even have a significant effect if native LPS was used in the in vitro MS experiments?

3) Lines 295-302: The authors report that LptB2FGC binds ReLPS with ~2-fold higher affinity than LptB2FG. They state that these results suggest "that LptC contributes to high-affinity substrate binding. The increase in affinity may result from either a decreased off-rate of LPS from the central cavity or increased binding to periplasmic domains of the transporter. Since previous structural studies have linked the presence of LptC's TM helix with lower LPS occupancy in the central cavity²⁸⁻³¹, this explanation with greater engagement of periplasmic domains, is more consistent with our data." I am not sure what "greater engagement of periplasmic domains" means. I presume it refers to "increased binding to periplasmic domains of the transporter". There is no data supporting that, in their experimental system, LPS is bound to the periplasmic domains of LptF, LptG, or even LptC. Was ATP added to cause LPS extraction and movement to the periplasmic domains as the authors imply? Can the authors rule out that the higher affinity of LptB2FGC for ReLPS is not an artifact of using *E. coli* LPS instead of the native *Pseudomonas* LPS? LPS from *Pseudomonas* has 5 fatty acyl chains, while *E. coli* LPS has 6. Since the LptC helix makes the cavity larger, it might increase binding to the *E. coli* LPS. Their conclusion would have to be supported with data using *Pseudomonas* LPS.

4) Based on MD simulations, the authors highlight 17 residues in LptF and 21 in LptG as relevant in lipid binding. They do not test the relevance of any of these residues but state: "Some of these residues, including R30, L62 (LptF) and R132 (LptG), are functionally validated: previous studies have shown that their mutation impairs bacterial viability". How many is "some"? 3 out of 38 residues? Were any of them tested in *Pseudomonas*? The authors could use the complementation system shown in Fig. S3 to test the importance of residues in *Pseudomonas*.

Minor issues:

- Lines 71-72: "an important role in regulating the ATPase activity to allow higher LPS extraction efficiency of the protein complex" is missing the beginning of the sentence.

- Figure 2d: Is this structure from LptB2FG-I? It should be indicated. Where in the complex is the cross-section shown in Fig. 2d? Is it at the predicted membrane-periplasmic interface? If the cross-section is of LptB2FG-I, please add markings indicating the plane of the cross-section shown in Fig. 2d. Similar issue with Fig. S7a and b. Please mark point of cross-section in the structure of the complex on the left. Also modify the figure legend in Fig. 7 to indicate structures on the right represent a cross-section.

- Lines 236-237: "resulting in an interface that resembles a heterodimer rather than a heterotrimer." ??

- Lines 191 and 258: "closed-up" should be "close-up".

- Lines 244-248: "A comparison of our cryo-EM structure of LptB2FGC-I with the previously reported *V. cholerae* LptB2FGC (6MJP), reveals a significant conformational change, with a RMSD of 5.57 Å across 845 C α atoms, indicating a significant conformational difference between these two structures. These differences suggest species-specific adaptations and further underscore the structural plasticity of the LptB2FGC complex." I do not understand why the authors suggest that the conformational change is the result of differences between species as opposed to stemming from the fact that the *Vibrio* structure has the TM helix of LptC in interface I, while the cryo-EM structure of LptB2FGC-I does not.

-Lines 269-271: "This affinity difference suggests that the transporter has adapted to recognise native *P. aeruginosa* LPS more tightly, likely due to subtle differences in the lipid A or core oligosaccharide structures between species." A supplemental figure showing the two LPS structures would be very helpful to readers and should be included.

-Lines 353-354: Citations are needed for: "particularly under conditions where cardiolipin is enriched, such as during stress responses, antibiotic treatment, or stationary phase".

-Figure 6: Please add "F" or "G" subscript/superscript to TM labels on structures on right to indicate whether they belong to LptF or LptG, respectively.

-Discussion: I am confused by this section: "It is interesting to note that, although LPS-bound, the recent *A. baylyi* LptB2FG structure (8FRM) adopts a closed conformation at Interface 2, which is similar to our substrate-free *P. aeruginosa* structure. The structural analysis suggests that *A. baylyi* LptB2FG adopts a conformation in which an LPS molecule has already entered the central cavity through interface 1, but the transmembrane helix of LptC (TMC) has not yet inserted. We therefore

propose this configuration as between our apo “resting” conformation and the TMC-capped form like observed in *V. cholerae* LptB2FGC.” Are the authors implying that LPS enters the cavity first and then LptC helix associates with the transporter? Such a model is very different from the current model and would imply that the LptC helix associates with the cavity after LPS enters it and then would have to leave again for LPS extraction. What data supports this model? Lines 446-447: “provides the molecular basis for LPS recognition and transport” should be “provides molecular basis for LPS recognition and transport”.

REVIEWER COMMENTS

Reviewer #1 (Remarks to the Author):

The authors have provided a satisfactory rebuttal to my prior criticisms. I'm grateful to see that my comments proved to be useful for the authors in providing a much more interesting revised manuscript. I should mention that the statement in lines 287-288 is redundant considering that the same point was made in lines 282-283.

We thank the reviewer for the positive evaluation. We agree that there is a redundancy and have now modified the text accordingly (lines 279-286, main text):

*“To dissect the specific features of LPS required for recognition by the LptB₂FG(C) complex, we performed native MS experiments using defined LPS analogues: Re-LPS, lipid A, and 4'-monophosphoryl lipid A (MPLA) (Figure 4a). These compounds differ progressively in their structural complexity: Re-LPS contains lipid A and two Kdo (3-deoxy-D-manno-oct-2-ulosonic acid) sugars; lipid A lacks the Kdo moieties; and MPLA is missing both Kdo and the phosphate group at position 1 of the glucosamine backbone. Only Re-LPS has been used previously to investigate Lpt system functionality^{17,47,48}. Because of the lack of commercial availability of these variants from *P. aeruginosa*, we used *E. coli* versions for our study.”*

Finally, regarding my query in Reviewer 1 point 4, the authors reply states "In theory, the phosphate-to-phosphate distance (~10 Å for cardiolipin versus ~6 Å for lipid IVA) can be distinguished in cryo-EM when local resolution is better than 3 Å and the lipid is well ordered." However, I ask the authors to consider that their numbers can't be correct as the phosphate groups in cardiolipin are closer together, while those of lipid A are farther apart. The reference I was citing is extended data Figure 9 in Clairfeuille T, et al. Structure of the essential inner membrane lipopolysaccharide-PbgA complex. Nature. (2020) 584: 479-483. This paper resolved some previous errors that had been made by structural biologists who failed to adequately distinguish between cardiolipin and lipid A. Since the present authors are making the same distinctions, they should be clear about what it is that they are comparing. The relevant link is:

<https://www.nature.com/articles/s41586-020-2597-x/figures/13>

We thank the reviewer for drawing attention to this important correction and for referencing Extended Data Figure 9 from Clairfeuille et al. (2020). We acknowledge the reviewer's point that the phosphate-to-phosphate distance in cardiolipin is shorter than in lipid A or lipid IVA, and we appreciate this clarification.

That said, we emphasise that our study focuses on ligand-free cryo-EM structures of LptB₂FGC, and we do not observe density consistent with either cardiolipin or lipid A in the central cavity. The lipid-like densities we do detect are confined to the periphery of the transmembrane domains and are consistent in shape and size with phosphatidylethanolamine (PE), which is expected based on our nanodisc reconstitution protocol. As such, our manuscript does not attempt to discriminate between lipid A and cardiolipin based on density interpretation, and the concern raised does not impact our structural conclusions.

Reviewer #2 (Remarks to the Author):

The authors provided a very thorough response, and I have no further significant edits to suggest. I appreciate the detailed description of what manuscript edits were made in response to each point raised.

We thank the reviewer for the positive assessment and thoughtful feedback throughout the revision process. We're pleased that our responses and manuscript edits address the points raised.

Reviewer #3 (Remarks to the Author):

The authors have improved the manuscript. I appreciate the work the authors put into revising the manuscript, especially by adding complementation studies, including data using native *Pseudomonas* LPS, and clarifying some important information in the text. Although the manuscript has been improved, there are still significant issues to address to justify several of the authors' conclusions.

Main issues:

1) The revised manuscript now shows that the tagged LptB protein complements in *Pseudomonas*. Specifically, the authors tested if relevant LptB proteins encoded in plasmids support growth of an lptB conditional strain in the absence of inducer. While this is a significant improvement of the study, the function should be more thoroughly tested for the following reasons. Previous studies have shown that although *E. coli* LptB-His has ATPase activity (in vitro) and complements the deletion of the lptB gene, it is partially functional in cells. It is functional enough to maintain growth but leads to significant defects in LPS transport, as demonstrated by the fact that cells with LptB-His are permeable (sensitive) to antibiotics and lptB-His can have either positive or negative effects in combination with specific lpt mutations. Whether the same is true or not in *Pseudomonas* remains unknown, but Fig. S3 shows that even though the depletion strain expressing LptB-KL-6×His (used in this study) can grow without inducer, it grows significantly smaller colonies than a strain producing a His-less LptB-KL. They also show that a depletion strain producing LptB-6×His grows better than LptB-KL-6×His but colonies are still smaller than colonies producing LptB. These data suggest that LptB-KL-6×His protein used in this study is partially functional, likely with significant defects in LPS transport based on the growth data. To describe it as functional is not appropriate. The effect of LptB-KL-6×His on the *Pseudomonas* Lpt transporter function remains uncharacterized. Whether structures would be different with a fully functional LptB is unknown, but it is a concern since the ATPase controls movement of the ABC transporter.

We thank the reviewer for raising this point. We agree that the small-colony phenotype indicates that C-terminally His-tagged LptB variants are partially functional in vivo and should not be described as fully functional. We have therefore tempered the wording throughout to explicitly state "partially functional."

In the revised manuscript we now state (lines 160-166, main text):

"Functional assays demonstrated that the His-tagged variants successfully complemented the lptB conditional mutant, even though the colonies were slightly smaller than those of the control strains expressing non-tagged LptB. This indicates that the tagged LptB variants are at least partially functional in vivo in P. aeruginosa (Figure S3). Together, these findings confirm that our purified LptB-FG(C) complexes are enzymatically competent in vitro and that the tagged LptB variants support growth in complementation assays and we therefore used these preparations for the structural and biochemical analyses reported here."

Regarding the functionality of the *Pseudomonas* Lpt transporter and the influence of LptB-KL-6×His on its structure, we fully agree that we cannot exclude subtle structural differences relative to an untagged protein, nor that differences might emerge under transport-active, ATP-turnover conditions. Importantly, we are not inferring in-cell activity from our structures: all cryo-EM data were collected in nucleotide-free conditions and capture apo conformations, so ATP-driven transport states are not interpreted here; characterising potential tag-dependent differences under turnover will be an important goal for future studies.

2) The authors now report that *Pseudomonas* LPS binds better to the *Pseudomonas* Lpt complex than the *E. coli* LPS (Fig S12). Unfortunately, they did not provide quantitative data. I could not find information about where the *Pseudomonas* LPS came from or how they purified it (unless I missed it, please add), but if the lack of quantitative data results from the fact that the *Pseudomonas* LPS is a mixture, I do not understand why they did not then purify it from any of the LPS mutants reported in the literature that would yield a more homogenous preparation. The lack of quantification is especially relevant to the issue about the biological significance of cardiolipin binding, which remains unclear mainly because there is no evidence that the authors' findings are relevant in *Pseudomonas* cells. Would cardiolipin even have a significant effect if native LPS was used in the in vitro MS experiments?

We thank the reviewer for this constructive comment. As correctly noted, preparations of *Pseudomonas* LPS are inherently heterogeneous. Although several groups have isolated LPS from rough or mutant *P. aeruginosa* strains, these invariably consist of polydisperse mixtures of glycoforms differing in phosphorylation, acetylation, acylation state, and core composition (e.g., Knirel et al., *Eur J Biochem.* 2001, 268(17):4708-19; Kropinski et al., *Can. J. Microbiol.* 1979, 25(3):390-8; Choudhury et al., *Carbohydr. Res.* 2005, 340(18):2761-72; Mei et al., *Microbiol. Spectrum* 2023, 11(6):e0177323). To our knowledge, a truly monodisperse *P. aeruginosa* LPS preparation suitable for quantitative MS has not been achieved, and chemical synthesis of defined lipid A analogues is technically challenging and prohibitively costly.

In line with this, we were unable to obtain a homogeneous preparation and therefore proceeded with a commercially available *P. aeruginosa* LPS (Sigma Aldrich, catalogue no. L8643), which we now specify in the Methods (supplementary information, lines 85-86). Importantly, the degree of heterogeneity in this preparation is not substantially different from that reported for mutant-derived LPS.

As requested, we now include a quantitative comparison of *P. aeruginosa* LPS and *E. coli* LPS binding (Figure S12, see below and in supplementary information, lines 379-388.). We quantified the different LPS-bound species by integrating the area under the curve for each peak — a widely accepted approach, especially when dealing with spectra exhibiting broad features. Nonetheless, explicitly note that these values should be regarded as semi-quantitative because of LPS heterogeneity.

Updated Figure S12. Pa- LptB₂FG binds its native LPS with higher affinity than *E. coli* LPS. Native mass spectra of Pa-LptB₂FG in presence of *P. aeruginosa* LPS (top) and *E. coli* LPS (bottom). 0.5 μ L of LPS at 5 mg/ml was added to 10 μ L of purified LptB₂FG at 3 μ M concentration. Since LPS is a heterogeneous mixture, a number of adduct peaks are observed in both cases. The bar chart on right shows the relative quantification, suggesting that LPS binding is much higher in the case of Pa-LPS compared to Ec-LPS. We explicitly caution that these values are semi-quantitative because of LPS heterogeneity.

Regarding cardiolipin, we agree that its physiological significance must be interpreted in the context of native *P. aeruginosa* LPS. Native LPS likely competes more effectively than *E. coli* Re-LPS. The limitation in obtaining a homogeneous preparation currently prevents testing this directly. The broader peak distributions in the presence of *P. aeruginosa* LPS also make additional binding experiments technically challenging and difficult to quantify. Nevertheless, our native MS experiments clearly show that cardiolipin binds directly to the Pa-LptB₂FGC complex and displaces Re-LPS in a concentration-dependent manner (Fig. 5b, S15–S16). These data provide the first direct evidence that cardiolipin can engage the Pa-Lpt transporter and modulate LPS binding. Moreover, under conditions of cellular stress, stationary phase and biofilm formation, cardiolipin levels are known to become significantly enriched in the inner membrane (see ref. 53-58 in the main text) and it might effectively compete for transient occupancy of the transporter's binding site, even against a higher-affinity substrate like Pa-LPS. That noted, independent genetic/physiological data support a functional role for cardiolipin in LPS transport: cardiolipin promotes transport across the inner membrane and can mitigate the transport deficit of under-acylated lipid A (e.g., *lpxM* contexts), providing in vivo plausibility for cardiolipin–Lpt interactions we observe biochemically (ref 51).

We have revised the manuscript to explicitly note that our competition experiments were performed with a model LPS substrate (Re-LPS) and thus represent a proof-of-concept rather than a definitive demonstration of in vivo function. We further acknowledge that the precise contribution of cardiolipin in the presence of native Pa-LPS remains an open question for future studies, particularly under physiological conditions (e.g., stress, stationary phase, biofilm formation) where cardiolipin levels are known to be enriched. We now add (lines 439-446, main text):

“Our lipid-binding studies further revealed that, beyond LPS, the transporter can accommodate cardiolipin (CDL), a lipid previously implicated in MsbA-mediated LPS transport⁵¹. The ability of CDL to bind and displace LPS (with a model LPS) in a concentration-

dependent manner, combined with MD simulations showing overlapping contact residues, raises the possibility that non-LPS lipids such as CDL might transiently interact with or regulate the transporter under specific cellular conditions (e.g., stress, membrane remodeling)⁵³⁻⁵⁸. This proof-of-concept observation adds a layer of physiological relevance to the transporter's substrate selectivity and suggests that LptB₂FGC may function within a lipid-rich regulatory environment, which remains to be tested in future studies.”

3) Lines 295-302: The authors report that LptB₂FGC binds ReLPS with ~2-fold higher affinity than LptB₂FG. They state that these results suggest “that LptC contributes to high-affinity substrate binding. The increase in affinity may result from either a decreased off-rate of LPS from the central cavity or increased binding to periplasmic domains of the transporter. Since previous structural studies have linked the presence of LptC’s TM helix with lower LPS occupancy in the central cavity²⁸⁻³¹, this explanation with greater engagement of periplasmic domains, is more consistent with our data.” I am not sure what “greater engagement of periplasmic domains” means. I presume it refers to “increased binding to periplasmic domains of the transporter”. There is no data supporting that, in their experimental system, LPS is bound to the periplasmic domains of LptF, LptG, or even LptC. Was ATP added to cause LPS extraction and movement to the periplasmic domains as the authors imply? Can the authors rule out that the higher affinity of LptB₂FGC for ReLPS is not an artifact of using *E. coli* LPS instead of the native *Pseudomonas* LPS? LPS from *Pseudomonas* has 5 fatty acyl chains, while *E. coli* LPS has 6. Since the LptC helix makes the cavity larger, it might increase binding to the *E. coli* LPS. Their conclusion would have to be supported with data using *Pseudomonas* LPS.

We thank the reviewer for pointing out this ambiguity. We agree that our original wording was unclear. Our equilibrium measurements show higher fractional LPS occupancy for LptB₂FGC than LptB₂FG under nucleotide-free conditions. This is consistent with a kinetic retention effect in which LptC lowers the effective dissociation rate (k_{off}) and/or increases the effective association rate (k_{on}) by gating lipid entry. Because our data report equilibrium occupancies, we cannot distinguish between changes in k_{on} and k_{off} here; accordingly, we present the reduced-off-rate explanation as a working hypothesis aligned with prior structural models of LptC-mediated gate opening. While we do not infer periplasmic-domain engagement from our assays, we cannot completely exclude additional interactions under our conditions because the exact TM-helix position is unknown. We have revised the relevant sentence in the manuscript accordingly to make this explicit in lines 296-303 of main text:

“The increase in affinity may result from either a decreased off-rate of LPS from the central cavity or increased binding to periplasmic domains of the transporter. Because our data are equilibrium occupancies, we cannot distinguish between changes in association and dissociation rates here; accordingly, we present the reduced-off-rate explanation as a working hypothesis aligned with prior structural models of LptC-mediated gate opening²⁸⁻³¹. While we do not infer periplasmic-domain engagement from our assays, we cannot completely exclude additional interactions under our conditions because the exact TM-helix position is unknown.”

The reviewer also asked whether the ~2 fold effect could be an artefact of using *E. coli* Re-LPS (hexa-acyl lipid A) rather than native Pa-LPS (often penta-acylated under lab conditions, though hexa-/hepta-acyl forms also occur in clinical isolates). To probe acylation-state generality, we re-analysed pre-delipidation native-MS datasets from ClearColi-expressed complexes, where lipid IVA (tetra-acyl) is the sole LPS-related species. In those data, LptB₂FG was predominantly apo (~60%), whereas LptB₂FGC showed markedly increased lipid IVA occupancy (~80% with 2× adducts), indicating that the LptC-dependent enhancement

of lipid occupancy extends to a tetra-acyl scaffold. While indirect, this argues against a hexa-acyl-specific artefact. See below.

Response only figure: Native mass spectra of PaLptB2FG(C) with lipid IVA. The relative quantification of lipid IVA shows significant increase in binding in the presence of LptC (bar chart on right).

4) Based on MD simulations, the authors highlight 17 residues in LptF and 21 in LptG as relevant in lipid binding. They do not test the relevance of any of these residues but state: “Some of these residues, including R30, L62 (LptF) and R132 (LptG), are functionally validated: previous studies have shown that their mutation impairs bacterial viability”. How many is “some”? 3 out of 38 residues? Were any of them tested in *Pseudomonas*? The authors could use the complementation system shown in Fig. S3 to test the importance of residues in *Pseudomonas*.

We clarified in the Results (lines 376-382) that “among the 38 cavity-contact residues highlighted by MD (17 LptF; 21 LptG), prior functional data exist for a subset of positions in enterobacteria (although not directly tested in *P. aeruginosa*), most notably hydrophobic/charged residues lining the cavity whose mutation compromises Lpt function or growth (e.g., LptF I25/R30/L62; LptG L26/R59/I70/R132/V308/F312). These studies collectively support the idea that the central cavity residues we observe contacting acyl chains in MD are functionally important, although numbering and exact identities vary by species/construct.”

Our study reports apo-state cryo-EM structures with no LPS density and does not include an LPS-bound structure or a mutational analysis; a systematic functional validation is therefore beyond the scope of this work. The complementation assay in Fig. S3 was designed to validate LptB variants for purification and in vivo functionality. Establishing an analogous complementation system to the essential membrane subunits LptF and LptG would require substantial new strain engineering and several months of work. This is outside the scope of the present study and is reserved for future work.

Minor issues:

- Lines 71-72: “an important role in regulating the ATPase activity to allow higher LPS extraction efficiency of the protein complex” is missing the beginning of the sentence.

This is now fixed.

- Figure 2d: Is this structure from LptB2FG-I? It should be indicated. Where in the complex is the cross-section shown in Fig. 2d? Is it at the predicted membrane-periplasmic interface? If the cross-section is of LptB2FG-I, please add markings indicating the plane of the cross-section shown in Fig. 2d. Similar issue with Fig. S7a and b. Please mark point of cross-section in the structure of the complex on the left. Also modify the figure legend in Fig. S7 to indicate structures on the right represent a cross-section.

Thank you for pointing that out. It is LptB2FG-I, and we have put this in the figure legends. We have added mark points in these figures.

Main text, line 198 and supplementary information, lines 347-351.

- Lines 236-237: "resulting in an interface that resembles a heterodimer rather than a heterotrimer." ??

We thank the reviewer for highlighting this unclear statement. We intended to describe how, in the *V. cholerae* structure, the LptG periplasmic β -jellyroll exhibits minimal interaction with LptC, in contrast to the more prominent LptC–LptF interface. We acknowledge that the original wording was confusing and have removed the sentence for clarity.

- Lines 191 and 258: "closed-up" should be "close-up".

This is now fixed.

- Lines 244-248: "A comparison of our cryo-EM structure of LptB2FGC-I with the previously reported *V. cholerae* LptB2FGC (6MJP), reveals a significant conformational change, with a RMSD of 5.57 Å across 845 C α atoms, indicating a significant conformational difference between these two structures. These differences suggest species-specific adaptations and further underscore the structural plasticity of the LptB2FGC complex." I do not understand why the authors suggest that the conformational change is the result of differences between species as opposed to stemming from the fact that the *Vibrio* structure has the TM helix of LptC in interface I, while the cryo-EM structure of LptB2FGC-I does not.

We thank the reviewer for this insightful comment. To clarify this point, we have revised the manuscript text to read (lines 245-247, main text):

*"A comparison of our cryo-EM structure of LptB \square FGC-I with the previously reported *V. cholerae* LptB \square FGC (6MJP) reveals a significant conformational difference (RMSD 5.57 Å across 845 C α atoms), which may reflect distinct functional states."*

-Lines 269-271: "This affinity difference suggests that the transporter has adapted to recognise native *P. aeruginosa* LPS more tightly, likely due to subtle differences in the lipid A or core oligosaccharide structures between species." A supplemental figure showing the two LPS structures would be very helpful to readers and should be included.

Thank you for this suggestion. We have now added a supplemental figure (Figure S13) showing two LPS structures. Lines 389-392, supplementary information.

-Lines 353-354: Citations are needed for: "particularly under conditions where cardiolipin is enriched, such as during stress responses, antibiotic treatment, or stationary phase".

References have now been added.

-Figure 6: Please add "F" or "G" subscript/superscript to TM labels on structures on right to indicate whether they belong to LptF or LptG, respectively.

Done.

-Discussion: I am confused by this section: “It is interesting to note that, although LPS-bound, the recent *A. baylyi* LptB2FG structure (8FRM) adopts a closed conformation at Interface 2, which is similar to our substrate-free *P. aeruginosa* structure. The structural analysis suggests that *A. baylyi* LptB2FG adopts a conformation in which an LPS molecule has already entered the central cavity through interface 1, but the transmembrane helix of LptC (TMc) has not yet inserted. We therefore propose this configuration as between our apo “resting” conformation and the TMc-capped form like observed in *V. cholerae* LptB2FGC.” Are the authors implying that LPS enters the cavity first and then LptC helix associates with the transporter? Such a model is very different from the current model and would imply that the LptC helix associates with the cavity after LPS enters it and then would have to leave again for LPS extraction. What data supports this model?

We thank the reviewer for this comment. The current working model suggests that the LptC transmembrane helix (TMc) associates with the cavity prior to LPS entry and later dissociate to allow LPS extraction. However, our apo structure of *P. aeruginosa* LptB□FGC displays an open interface I without TMc insertion. This observation led us to consider an alternative model in which LPS might enter the cavity before TMc insertion. The recent *A. baylyi* LptB□FG–LPS structure (PDB: 8FRM), although LPS-bound, shows a closed interface II. While these findings raise the possibility of a different sequence of events, we cannot rule out that TMc may still transiently insert into the interface to facilitate LPS capture and release. Additionally, we acknowledge that the order and coordination of TMc insertion and substrate binding may be species dependent. To avoid overinterpretation, we have revised the text to more cautiously state (in main text, lines 429-432):

“It is interesting to note that, although LPS-bound, the recent A. baylyi LptB□FG structure (PDB: 8FRM) adopts a closed conformation at interface II, similar to our substrate-free P. aeruginosa structure. Structural analysis suggests that this A. baylyi conformation captures a state in which LPS has entered the cavity through interface I.”

Lines 446-447: “provides the molecular basis for LPS recognition and transport” should be “provides molecular basis for LPS recognition and transport”.

This is now fixed.